# Sequential Preference Ranking for Efficient Reinforcement Learning from Human Feedback

**Minyoung Hwang**[1], **Gunmin Lee**[1], **Hogun Kee**[1], **Chanwoo Kim**[2],
**Kyungjae Lee**[2*], **Songhwai Oh**[1*]
{minyoung.hwang, gunmin.lee, hogun.kee}@rllab.snu.ac.kr,
jenior97@cau.ac.kr, kyungjae.lee@ai.cau.ac.kr, songhwai@snu.ac.kr

[1]Electrical and Computer Engineering and ASRI
Seoul National University
Seoul, 08826, Korea

[2]Department of Artificial Intelligence
Chung-Ang University
Seoul 06974, Korea

## Abstract

Reinforcement learning from human feedback (RLHF) alleviates the problem of designing a task-specific reward function in reinforcement learning by learning it from human preference. However, existing RLHF models are considered inefficient as they produce only a single preference data from each human feedback. To tackle this problem, we propose a novel RLHF framework called SeqRank, that uses sequential preference ranking to enhance the feedback efficiency. Our method samples trajectories in a sequential manner by iteratively selecting a defender from the set of previously chosen trajectories $\mathcal{K}$ and a challenger from the set of unchosen trajectories $\mathcal{U} \setminus \mathcal{K}$, where $\mathcal{U}$ is the replay buffer. We propose two trajectory comparison methods with different defender sampling strategies: (1) sequential pairwise comparison that selects the most recent trajectory and (2) root pairwise comparison that selects the most preferred trajectory from $\mathcal{K}$. We construct a data structure and rank trajectories by preference to augment additional queries. The proposed method results in at least $39.2\%$ higher average feedback efficiency than the baseline and also achieves a balance between feedback efficiency and data dependency. We examine the convergence of the empirical risk and the generalization bound of the reward model with Rademacher complexity. While both trajectory comparison methods outperform conventional pairwise comparison, root pairwise comparison improves the average reward in locomotion tasks and the average success rate in manipulation tasks by $29.0\%$ and $25.0\%$, respectively. Project page: https://rllab-snu.github.io/projects/SeqRank

## 1 Introduction

Designing a suitable reward function in reinforcement learning often requires task-specific prior knowledge [1] and sufficient time to design the reward function to capture the true task objective. Without this effort, the agent may be easily led to suboptimal policies. These limitations motivate the use of reinforcement learning from human feedback (RLHF), which can directly learn from human's preferences without the need for a hand-crafted reward function [1]. A conventional way to learn a reward function in RLHF is pairwise comparison: querying human preference between two different trajectories as described in Figure 1. Despite the significant success of RLHF, conventional pairwise comparison requires a human to remember at least two trajectories to determine a single preference.

---

*Corresponding Authors

37th Conference on Neural Information Processing Systems (NeurIPS 2023).

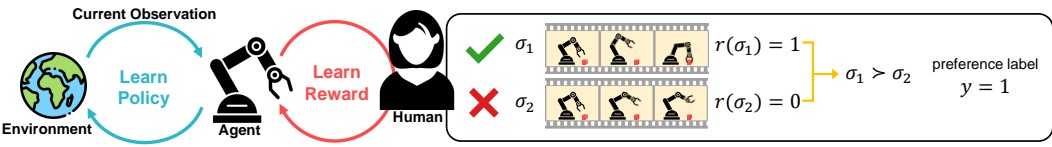

**Figure 1: Overview.** Reinforcement learning from human feedback captures human preference against agent trajectories. While the agent learns the control policy by interacting with the environment, it also learns the reward function from human feedback, given as binary labels for trajectory pairs.

In this regard, it is essential to develop an efficient comparison method that can obtain multiple preference data from a single human feedback, thus reducing a human's labeling effort. In this paper, we define the concept of *feedback efficiency* as *the ratio between the number of trajectory pairs over the number of feedbacks*, where the feedback efficiency of conventional pairwise comparison is set at a standardized level. We aim to develop a novel RLHF framework with greater feedback efficiency than existing methods.

The key idea of our approach is to utilize the preference relationships of the previous trajectory pairs. Bringing the nature of transitivity in human preferences [2], given three trajectories, $\sigma_i, \sigma_j$, and $\sigma_k$, a human with preference over $\sigma_i$ to $\sigma_j$ and $\sigma_j$ to $\sigma_k$ also prefers $\sigma_i$ over $\sigma_k$. Hence, if we maintain the history of preference relationships, we can easily augment preference data without additional human feedback. In the proposed method, the comparison process maintains two sets of trajectories. One collects previously compared trajectories, and the other includes remaining trajectories that have not yet been compared. We iteratively select and compare a *defender* trajectory from the set of previously chosen trajectories and a *challenger* trajectory from the set of unchosen trajectories. In this paper, we propose two trajectory comparison methods: (1) sequential pairwise comparison that selects the most recently sampled trajectory and (2) root pairwise comparison that selects the previously most preferred trajectory as the defender. For both methods, the agent can augment additional preference data from the comparison results due to the transitivity.

We show that the proposed sequential preference ranking substantially improves the feedback efficiency and significantly outperforms conventional pairwise comparison. We prove that the proposed method converges faster for the empirical risk of the reward model than conventional pairwise comparison. One drawback of the proposed method is that it generates dependencies within the original and the augmented preference data, which are both used to train the reward model. Still, we prove the generalization bound of the reward model with Rademacher complexity converges with rates of $\mathcal{O}(\sqrt{(\ln(T))^2/T})$ and $\mathcal{O}(\sqrt{\ln(T)/T})$ for sequential and root pairwise comparison, respectively, where $T$ is the number of iterations. We demonstrate the effectiveness of the proposed method in both locomotion tasks from Deepmind Control Suite (DMControl) [3] and manipulation tasks from Meta-World [4]. Among the proposed two trajectory comparison methods, root pairwise comparison shows the highest performance in all tasks, which is improved by $29.0\%$ and $25.0\%$ against the pairwise baseline in locomotion and manipulation tasks, respectively. Additional experiments using real human feedback validate the effectiveness of the proposed method, showing significant improvements in true reward, sample efficiency, feedback efficiency, and user satisfaction. Furthermore, our method successfully outperforms the baseline in terms of success rates in controlling a real UR-5 robot for a block placing task, demonstrating its practical applicability in the real world.

The main contributions of this paper are as follows:

- We propose a novel RLHF framework that utilizes sequential preference ranking to enhance human feedback efficiency. We prove the proposed sequential and root pairwise comparison substantially improve the average feedback efficiency by $39.2\%$ and $100\%$, respectively. An increase in feedback efficiency speeds up the estimation of the reward function.

- We derive the convergence rates of the empirical risk and the generalization bound of the reward model with Rademacher complexity using the proposed sequential and root pairwise comparison. We address the trade-off between feedback efficiency and data dependency required for successful reward learning.

- We empirically show that prioritizing the feedback efficiency is significantly important by evaluating in simulation and real-world environments. Both sequential and root pairwise

comparison outperform conventional pairwise comparison on average. Root pairwise comparison shows the most substantial improvement against the baseline by 29.0% and 25.0% in DMControl locomotion and Meta-World manipulation tasks, respectively.

## 2 Related Work

**Reinforcement Learning from Human Feedback.** A common approach in reinforcement learning from human feedback, i.e., preference-based reinforcement learning, is to learn the reward function from explicit feedback, such as ratings or comparisons provided by human experts. Christiano *et al.* [5] present the human-in-the-loop reinforcement learning framework for robotic tasks. Lee *et al.* (2021a) [6] propose a benchmark that introduces simulated human teachers with various irrationalities. Recent work [7–10] introduce efficient RLHF algorithms. Lee *et al.* (2021b) [7] propose a method that improves Christiano *et al.* [5] via unsupervised pre-training and off-policy RL with relabeling. Park *et al.* [8] suggest a method that uses semi-supervised learning and data augmentation. Liang *et al.* [9] show that exploration based on uncertainty in learned reward functions helps to improve the performance with fewer samples and feedbacks. Liu *et al.* [10] propose a data-efficient RLHF framework incorporating bi-level optimization for reward and policy learning.

However, most of the previous studies still achieve low feedback efficiency as they do not take into account the nature of ranking through trajectory comparisons. Several recent work [11–13] suggest querying a complete ranking among multiple ($\geq 3$) trajectories, but this assigns the responsibility of ranking trajectories to humans, not robots. Requesting a human to remember multiple trajectories at once and provide best-of-multiple feedback [11] is also challenging due to human uncertainty [14]. To solve this challenge, we propose sequential preference ranking, which allows the agent to automatically rank trajectories while a human only provides pairwise comparisons based on one's preference. The key idea is to perceive piecewise ranks instead of ranking the trajectories as a whole.

**Generalization Bounds for Learning Under Graph-Dependence.** While data augmentation via sequential preference ranking can improve the efficiency of reward learning, this can lead to dependencies within the training data. However, traditional learning theories are based on the assumption that data are identically and independently distributed (i.i.d.). Hence, to analyze the generalization bound of the proposed method, we investigate the generalization bounds for learning with data dependencies. A common approach to consider data dependency is to construct a graph that characterizes the dependency relationship within the data [15]. Janson *et al.* [16] establish that the probability of the sum of graph-dependent random variables deviating from its expected value is bounded based on the fractional chromatic number of the dependency graph, by extending Hoeffding's inequality. Usunier *et al.* [17] present a concentration inequality by extending fractional Rademacher complexity, and prove generalization bounds for binary classification over interdependent data. Zhang *et al.* (2019) [18] propose McDiarmid-type concentration inequalities for Lipschitz functions of graph-dependent random variables and show that concentration relies on the forest complexity of the dependency graph. In this paper, we derive the generalization bound of the reward model with Rademacher complexity in terms of the maximum degree of the dependency graph and the number of training data, which includes augmented preference queries and labels.

## 3 Learning the Reward Function from Human Preferences

Following prior work [5, 7, 10, 19], we use a deterministic human model that provides preference as a binary feedback over a pair of trajectory segments. We define the true preference label $y$ for a pair of segments $(\sigma_i, \sigma_j)$ as follows:

$$y = \begin{cases} 1, & \text{if } \sum_{t=1}^{H} r(s_t^i, a_t^i) > \sum_{t=1}^{H} r(s_t^j, a_t^j) \\ 0, & \text{otherwise,} \end{cases} \tag{1}$$

where $r$ is the true reward function of a state-action pair $(s_t^i, a_t^i)$ in $\sigma_i$ at timestep $t$ and $H$ is the length of a segment. The true reward of a trajectory segment $\sigma = (s_1, a_1, ..., s_H, a_H)$ can be written as $r(\sigma) = \sum_t r(s_t, a_t)$. Similar to prior work [13, 20–22], we implement our algorithm with a linearly parameterized preference model. Assume that there exist a nonlinear trajectory encoder $\phi : \mathcal{T} \rightarrow \mathbb{R}^d$ and a parameter $\theta^* \in \mathbb{R}^d$ that represents the true reward function as $r(\sigma) := \theta^{*\intercal}\phi(\sigma)$, where $d$ is the dimension of $\theta^*$ and $\mathcal{T}$ is a set of trajectory

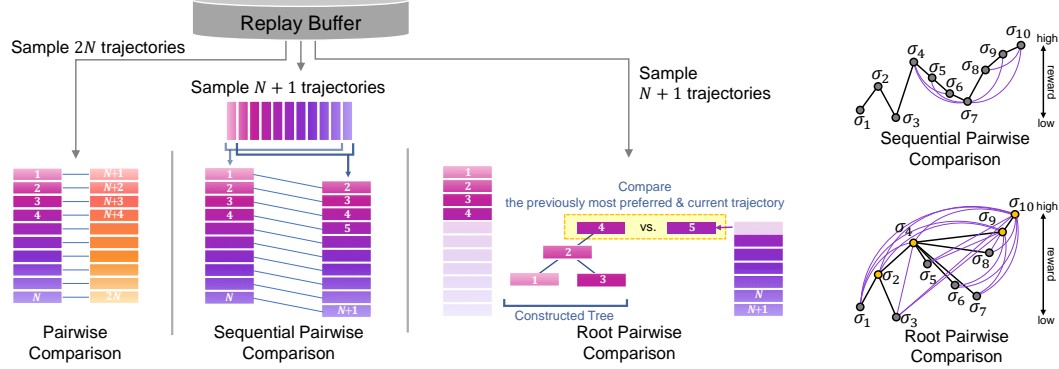

**(a)** Three trajectory comparison methods.

**(b)** Sequential vs Root Pairwise Comparison

**Figure 2: Trajectory comparison methods.** (a) All three methods query a human to provide $N$ feedback. Pairwise comparison samples $2N$ trajectories to obtain $N$ feedback, while sequential or root pairwise comparison samples $N+1$ trajectories. Despite sampling fewer trajectories, sequential or root pairwise comparison shows higher feedback efficiency than pairwise comparison by adopting sequential preference ranking. (b) Gray nodes illustrate fixed-length trajectory segments sampled from the replay buffer. Suppose the reward values for segments $\sigma_1, ..., \sigma_{10}$ are $2, 5, 1, 8, 6, 4, 3, 7, 9, 10$, respectively. Black lines indicate actual pairs that receive true preference labels from human feedback. The upper node for each black line represents the preferred trajectory. For root pairwise comparison, orange nodes ⬤ describe non-leaf nodes in the tree. Using sequential or root pairwise comparison, the agent can obtain augmented labels for non-adjacent pairs illustrated with purple lines.

segments. We estimate the reward using a learnable reward function $\hat{r}_\theta$, parameterized by $\theta$. Then, the estimated reward of $\sigma$ can be written as $\hat{r}_\theta(\sigma) = \sum_t \hat{r}_\theta(s_t, a_t) = \theta^\mathsf{T} \phi(\sigma)$. Preference prediction is also interpreted as a binary classification problem with a class probability $P(y = 1|(\sigma_i, \sigma_j); \theta^*) = P_{\theta^*}[\sigma_i \succ \sigma_j] = \exp(\sum_t r(s_t^i, a_t^i))/\sum_{k \in \{i,j\}} \exp(\sum_t r(s_t^k, a_t^k))$ based on the Bradley-Terry model [23], where $\sigma_i \succ \sigma_j$ denotes that the $i^{th}$ trajectory segment is preferable to the $j^{th}$ trajectory segment. For training the reward model, we use $P(\hat{y} = 1|(\sigma_i, \sigma_j); \theta) = P_\theta[\sigma_i \succ \sigma_j] = \exp(\sum_t \hat{r}_\theta(s_t^i, a_t^i))/\sum_{k \in \{i,j\}} \exp(\sum_t \hat{r}_\theta(s_t^k, a_t^k))$ to estimate the preference label $\hat{y}$ of $(\sigma_i, \sigma_j)$. If $P_\theta[\sigma_i \succ \sigma_j]$ is greater than $0.5$, $\hat{y}$ is $1$. Otherwise, $\hat{y}$ is $0$. Using cross-entropy loss, we define the preference loss function $L_{pref}$ as follows:

$$L_{pref} = -\mathbb{E}_{((\sigma_i, \sigma_j), y) \in \mathcal{D}}[y \log P_\theta[\sigma_i \succ \sigma_j] + (1 - y) \log P_\theta[\sigma_i \prec \sigma_j]], \tag{2}$$

where $\mathcal{D}$ is the set of sampled trajectory segment pairs and their corresponding true preference labels.

In traditional RLHF methods, a common approach to sampling a segment pair is to randomly sample each segment from the replay buffer, resulting in the selection of irrelevant pairs. This independent sampling approach does not take into account any inherent relationships between the non-adjacent segments, where two segments are considered adjacent if they are compared directly from a human. In contrast, the proposed method adopts a sequential sampling approach that enables indirect inference of the relationship between non-adjacent segments. The proposed method allows for more comprehensive understanding of the preferences among trajectory segments, capturing both direct and indirect relationships.

## 4 SeqRank

We present SeqRank, which samples trajectories and compares them in a sequential manner for efficient reinforcement learning from human feedback. We also extend our key component, sequential pairwise comparison, to root pairwise comparison, which not only compares trajectories sequentially but also constructs a tree structure to rank trajectories in a more efficient way. Figure 2 (a) illustrates the difference between three trajectory comparison methods including conventional pairwise comparison and the proposed sequential and root pairwise comparison. We demonstrate the feedback efficiency and convergence of the reward model for each method. Regarding the notations used in our paper, a summary of symbols and terms can be found in Table 1 in the supplementary material.

We first make the following assumptions:

**(Uniformness)** Every trajectory is sampled uniformly from the replay buffer.

**(Completeness)** For any two segments $\sigma_i$ and $\sigma_j$, "$\sigma_i \succ \sigma_j$ or $\sigma_i \prec \sigma_j$" is true.

**(Transitivity)** For any three segments $\sigma_i, \sigma_j$, and $\sigma_k$, if $\sigma_i \succ \sigma_j$ and $\sigma_j \succ \sigma_k$, then $\sigma_i \succ \sigma_k$.

Assuming $N$ feedback per reward update, the agent samples the $(n+1)^{th}$ defender and challenger from $\mathcal{K}_n = \{\sigma_1, ..., \sigma_n\}$ and $\mathcal{U} \setminus \mathcal{K}_n$, respectively, where $\mathcal{U}$ is the replay buffer. The defender becomes $\sigma_n$ for sequential pairwise comparison and $\arg\max_{\sigma \in \mathcal{K}_n} \theta^\mathsf{T}\phi(\sigma)$ for root pairwise comparison. After the $N^{th}$ query, the agent receives $N$ feedback in total. Figure 2 (b) illustrates the preference relationships among trajectory segments as a graph. Especially for root pairwise comparison, the challenger becomes a new root of the tree if it is preferred over the current root segment as described in Figure 2 (a). Otherwise, the challenger becomes a new child of the current root. The full procedure of our algorithm is provided in the supplementary material. Suppose $S_M$ is the total number of trajectory segment pairs obtained from $M$ trajectory segments. $M = 2N$ for conventional pairwise comparison, and $M = N + 1$ for sequential or root pairwise comparison. In the following section, we first demonstrate the feedback efficiency in terms of $M$ and substitute it with the function of $N$.

### 4.1 Feedback Efficiency of the Algorithm

From the uniformness assumption, suppose the agent compares $M$ trajectory segments with corresponding rewards as a random ordering of $1, 2, ..., M$. Based on the completeness and transitivity assumptions, the agent can augment non-adjacent segment pairs as illustrated in Figure 2 (b). Considering that the number of augmented pairs differs by the order of sampling trajectories with different rewards, $S_M$ becomes a random variable indicating the total number of pairs collected from $M$ trajectory segments. In this section, we demonstrate the average feedback efficiency of our method.

*Definition* **1.** *We define **feedback efficiency** as the ratio between the total number of trajectory pairs over the number of feedbacks. The expected feedback efficiency for $N$ human feedback is $\mathbb{E}[S_{N+1}]/N$ for sequential and root pairwise comparison.*

**Sequential Pairwise Comparison.** Following Definition 1, we use Lemma 1 to examine the expected number of segment pairs collected from sequential pairwise comparison.

*Theorem* **1.** *For any integer $M \geq 2$, the expected number of trajectory segment pairs collected from sequential pairwise comparison $a_M$ is approximately linear to $M$ as $|a_M/M - 1/(e - 2)| = o(1)$.*

*Lemma* **1.** *For any integer $M \geq 2$, $a_M$ can be induced from the following recurrence relation.*

$$a_M = \frac{\textit{(total number of pairs from all possible orderings of trajectories)}}{\textit{(number of all possible orderings of trajectories)}}$$
$$= 2 - 2/M! + \sum_{j=2}^{M-1} a_{M-j+1}(2j/(j+1)!) \tag{3}$$

By solving the recurrence relation stated in Lemma 1, we derive Theorem 1 that $a_M$ is approximately equal to a linear equation $\alpha(M - \gamma)$ with an error less than $\epsilon$, for any $M$ such that $M!$ is greater than $2\gamma/\epsilon$. $\alpha = 1/(e-2) \approxeq 1.392$ and $\gamma > 1$, respectively. By substituting a sufficiently large value of $M \geq 100$, $\gamma$ is calculated as $1.324$, and the expected feedback efficiency of sequential pairwise comparison becomes $1.392(M - 1.324)/(M - 1)$, which converges to $1.392$ as $M$ increases. Moreover, for any $M \geq 4$, the linear approximation error of $a_M$ is bounded by $0.003$. Detailed proofs of Lemma 1 and Theorem 1 are in the supplementary material.

Comparing the proposed sequential pairwise comparison with conventional pairwise comparison that requires $2N$ trajectory segments for $N$ human feedback, we observe that sequential pairwise comparison only requires $N + 1$ trajectory segments to query a human for $N$ feedback. Additionally, using the proposed sequential pairwise comparison, the expected number of augmented queries is given by $a_{N+1} - N \approxeq 0.392N - 0.451$.

**Extension to Root Pairwise Comparison.** We further extend sequential pairwise comparison to root pairwise comparison by exploiting a specific defender sampling strategy. The key idea is to compare the current segment with the most preferred one from the previously sampled segments. As described in Figure 2 (b), root pairwise comparison augments more pairs than sequential pairwise comparison by constructing a tree structure. Using root pairwise comparison, the average feedback efficiency is increased to two. Theorem 2 demonstrates the average feedback efficiency of our algorithm using root pairwise comparison. The detailed proof of Theorem 2 is in the supplementary material.

| Method | $M$ | Best | | Average | | Worst | |
|---|---|---|---|---|---|---|---|
| | | $p_N$ | $\eta$ | $p_N$ | $\eta$ | $p_N$ | $\eta$ |
| Pairwise | $2N$ | $N$ | $1$ | $N$ | $1$ | $N$ | $1$ |
| Sequential Pairwise | $N+1$ | $N(N+1)/2$ | $(N+1)/2$ | $1.392(N-0.324)$ | $1.392$ | $N$ | $1$ |
| Root Pairwise | $N+1$ | $N(N+1)/2$ | $(N+1)/2$ | $2(N+1-\sum_{n=1}^{N+1}\frac{1}{n})$ | $2$ | $N$ | $1$ |

Table 1: **Trajectory comparison methods.** We compare three trajectory comparison methods: pairwise, sequential pairwise, and root pairwise. $\eta$ denotes the feedback efficiency.

**Theorem 2.** *In root pairwise comparison, for any integer $M \geq 2$, the expected number of pairs $b_M$ can be denoted by*

$$b_M = 2(M - \sum_{n=1}^{M}(1/n)). \tag{4}$$

Considering that $\sum_{n=1}^{M}\frac{1}{n} < 1 + \int_1^M \frac{1}{x}dx = 1 + \ln M$, $(\sum_{n=1}^{M}\frac{1}{n})/(M-1)$ converges to 0 as $M$ increases. Therefore, root pairwise comparison achieves the expected feedback efficiency as $b_M/(M-1) \cong 2$. This implies that the proposed root pairwise comparison achieves higher feedback efficiency than sequential pairwise comparison on average.

Table 1 shows the comparison results of trajectory comparison methods when the number of human feedback is fixed to $N$. $M$ and $p_N$ denote the number of sampled trajectories at each reward update session and the expected number of trajectory pairs obtained from $N$ human feedback, i.e., $\mathbb{E}[S_N]$, respectively. For sequential and root pairwise comparison methods, $p_N$ is $a_{N+1}$ and $b_{N+1}$, respectively. The baseline method using conventional pairwise comparison samples $2N$ trajectories, which are approximately twice as many of sequential or root pairwise comparison methods. As shown in Table 1, sequential pairwise comparison achieves an average feedback efficiency of 1.392. Also, root pairwise comparison achieves the highest average feedback efficiency of 2.

## 4.2 Convergence of the Reward Model

**Notations.** Suppose $(\phi(\sigma_i)-\phi(\sigma_j))$ is the input and $\sigma_i \succ \sigma_j$ is the true label of a binary classification using a logistic regression model and a cross-entropy loss. Then, $P_\theta[\sigma_i \succ \sigma_j]$ can be interpreted as a logistic regression model as follows:

$$P_\theta[\sigma_i \succ \sigma_j] = \frac{1}{1 + \exp(\sum_t \hat{r}_\theta(s_t^j, a_t^j) - \hat{r}_\theta(s_t^i, a_t^i))} = \frac{1}{1 + \exp(-\theta^\mathsf{T}(\phi(\sigma_i) - \phi(\sigma_j)))}. \tag{5}$$

For simplicity, we define the training data for reward update as $\mathcal{D} = \{(x_k, y_k)\}$, where $x_k$ and $y_k$ denote the features of two trajectories included in the $k^{th}$ query and its corresponding preference label provided from a human, respectively. Note that if the $k^{th}$ query is $(\sigma_i, \sigma_j)$, $x_k = \phi(\sigma_i) - \phi(\sigma_j)$. We define the true risk of the reward model as $R(\theta) := \mathbb{E}[\ell(\theta^{*\mathsf{T}}(x), \theta^\mathsf{T}(x))]$ and the empirical risk of the reward model as $\hat{R}(\theta) := L_{pref} = (1/|\mathcal{D}|)\sum_{x_k \in \mathcal{D}} \ell(\theta^{*\mathsf{T}}(x_k), \theta^\mathsf{T}(x_k))$, where the cross-entropy loss and its minimizer are denoted as $\ell$ and $\theta^*$, respectively. Assume that $||\phi||_2$, $||\theta^*||_2$, and $||\nabla\theta||_2$ are bounded with constants $S$, $D$, and $Q$, respectively. Since RLHF involves an iterative process of policy and reward update, we distinguish the overall iteration $T$ that represents the global iterative process, and the local iterations of policy and reward update, $W$ and $U$, respectively. $S_{M,T}$ represents the cumulative sum of the number of training data of the reward model from iteration 1 to $T$, with $M$ segments compared per iteration. We introduce the concept of dependency graph and its maximum degree $\Delta_{M,t}$ at iteration $t \in [T] = \{1, 2, ..., T\}$ for further analysis, following previous work [15]. Detailed proofs for the following theorems, lemmas, and corollaries are in the supplementary material.

**High Probability Feedback Efficiency.** We find a high probability bound for $S_{M,T}$ in terms of the expected values of $S_M$ as described in Lemma 2. This implies that for sufficiently large $T$ and $N$, the feedback efficiency averaged across $T$ iterations is at least equal to $p_N/(2N)$ in the proposed method with high probability. For sequential and root pairwise comparison, the lower bounds become 0.696 and 1, respectively, while the feedback efficiency for pairwise comparison is fixed to 1.

**Lemma 2.** *For a sufficiently large $T > (\max_t \Delta_{M,t} + 1)(M-1)\ln(1/\delta)/(\beta^2 M)$ and $\delta \in (0, 1)$, with probability at least $1 - \delta$, $S_{M,T}$ is greater than $\beta T M/2$, where $\beta$ denotes the expected number of pairs obtained from $M$ segments divided by $M$.*

**Convergence Rate of the Empirical Risk.** From the completeness assumption, data in $\mathcal{D}$ should be linearly separable. According to prior work [24], the empirical risk converges with a rate of $\mathcal{O}(1/U)$. Let $n_B$ be the batch size of reward update. Then, after the global iteration $T$, $U = \lfloor S_{M,T}/n_B \rfloor$.

From Lemma 2, we prove that with probability at least $1 - \delta$ and a sufficiently large $T$, the empirical risk of the reward model converges at a rate of $\mathcal{O}(2n_B/(\beta TM))$. Therefore, higher $\beta$ results in faster convergence. While $(\beta, M)$ for pairwise comparison are fixed as $(0.5, 2N)$, the values for sequential and root pairwise comparison are approximated to $(1.392, N+1)$ and $(2, N+1)$, respectively. Based on our analysis, the reward model is likely to converge faster in the order of root pairwise, sequential pairwise, and pairwise comparison in terms of the global iteration $T$.

**Convergence Rate of the Generalization Bound.** Based on prior work [15, 25], we derive a generalization bound with Rademacher complexity for learning the reward model under graph-dependence in $\mathcal{D}$. Theorem 3 shows the generalization bound of the reward model. Detailed proof of the theorem and definitions of $F$ and $B$ are presented in the supplementary material.

**Theorem 3.** *Assume that a comparison algorithm generates a train set $\mathcal{D}$ of size $S_{M,T}$ with dependency graph $G$ after $T$ iteration. Then, for any $\delta \in (0, 1)$, with probability at least $1 - \delta$,*

$$\forall \theta \in \mathcal{H}, \ \ R(\theta) \le \hat{R}(\theta) + \sqrt{(\max_t \Delta_{M,t} + 1)/S_{M,T}} \cdot (2FD + 3B\sqrt{\ln(4/\delta)/2}), \tag{6}$$

*where $\mathcal{H}$ is the parameter space of $\theta$, and $F$ and $B$ are functions of $M$ with rates at $\mathcal{O}(M^2)$.*

Theorem 3 addresses a critical aspect of the proposed method by proving that even with dependent data, the empirical risk of the reward model converges to the true risk with probability at least $1 - \delta$. This high probability bound contributes to the practical analysis of general RLHF models with graph-dependent data such as rankings. Additionally, we consider the trade-off between feedback efficiency ($\propto S_{M,T}$) and data dependency ($\propto \Delta_{M,t}$). As shown in (6), lower $\Delta_{M,t}$ and larger $S_{M,T}$ lead to a tighter generalization bound of the reward model. However, data dependency increases in our settings as we augment more queries and enhance feedback efficiency. Hence, we demonstrate $\Delta_{M,t}$ for sequential and root pairwise comparison in Lemma 3 and 4, respectively.

**Lemma 3.** *Let $G$ be a dependency graph generated by sequential pairwise comparison with $M$ trajectory segments. Then, for any $\delta \in (0, 1)$, the maximum degree of $G$ is bounded as follows, with probability at least $1 - \delta$,*

$$\max_{t \in [T]} \Delta_{M,t} \le 2 + \ln(1 + M/(\ln(1/(1-\delta)))). \tag{7}$$

**Lemma 4.** *Let $G$ be a dependency graph generated by root pairwise comparison with $M$ trajectory segments. Then, for any $\delta \in (0, 1)$, the maximum degree of $G$ is bounded as follows, with probability at least $1 - \delta$,*

$$\max_{t \in [T]} \Delta_{M,t} \le M(2 - \delta) - 3. \tag{8}$$

**Corollary 3.1.** *For a fixed $M \ge 2$, the generalization bounds of the reward model for pairwise, sequential pairwise, and root pairwise comparison converge at rates of $\mathcal{O}(\sqrt{\ln(T)/T})$, $\mathcal{O}(\sqrt{(\ln(T))^2/T})$, and $\mathcal{O}(\sqrt{\ln(T)/T})$, respectively, with probability at least $1 - 1/T$.*

By applying Lemma 2, 3, and 4 to Theorem 3, we derive Corollary 3.1, which compares the generalization bounds of three methods. While generalization bounds of all methods are guaranteed to converge as $T$ increases, root pairwise comparison demonstrates a faster convergence rate than sequential pairwise comparison, and has the same convergence rate as pairwise comparison.

**Overall analysis of convergence.** To sum up, root pairwise comparison theoretically surpasses the baseline as it proves better and equal convergence rates of both metrics, while requiring a half of samples than pairwise comparison. Sequential pairwise comparison achieves faster convergence of the empirical risk but slower convergence of the generalization bound compared to pairwise comparison. From the perspective of reward model convergence, prioritizing the feedback efficiency to speed up the convergence of the empirical risk is more critical than considering the generalization error in worst-case scenarios. In Section 5, we empirically show the superiority of root pairwise comparison over sequential and pairwise comparison.

## 5 Experiments

We evaluate our method on robotic locomotion tasks from DeepMind Control Suite (DMControl) [3, 26] and robotic manipulation tasks from Meta-World [4]. We also demonstrate the proposed

| Task | # feedback | Oracle | Pairwise | Sequential Pairwise | Root Pairwise |
|---|---|---|---|---|---|
| Walker Walk | 0.4K | 957.3 $\pm$ 2.3 | 862.5 $\pm$ 85.3 | 883.7 $\pm$ 71.9 | **907.5** $\pm$ 66.2 |
| Cheetah Run | 0.2K | 886.9 $\pm$ 43.2 | 690.2 $\pm$ 93.2 | 715.8 $\pm$ 117.4 | **739.4** $\pm$ 100.8 |
| Quadruped Walk | 1K | 843.3 $\pm$ 247.3 | 353.5 $\pm$ 183.9 | 479.5 $\pm$ 234.7 | **728.8** $\pm$ 274.1 |
| Humanoid Walk | 40K | 272.5 $\pm$ 117.0 | 114.3 $\pm$ 80.5 | 141.3 $\pm$ 34.4 | **163.8** $\pm$ 71.6 |
| Hopper Hop | 4K | 273.5 $\pm$ 47.0 | 26.2 $\pm$ 37.0 | 37.3 $\pm$ 59.8 | **100.1** $\pm$ 70.7 |

**Table 2: Rewards after convergence in DMControl locomotion tasks.** Two elements in each cell denote the average value and standard deviation of rewards across runs with 10 random seeds.

| Task | # feedback | Oracle | Pairwise | Sequential Pairwise | Root Pairwise |
|---|---|---|---|---|---|
| Button Press | 10K | 99.3 $\pm$ 0.9 | 95.6 $\pm$ 7.6 | 97.0 $\pm$ 4.3 | **97.6** $\pm$ 5.6 |
| Door Open | 10K | 100.0 $\pm$ 0.0 | 77.9 $\pm$ 41.3 | 77.4 $\pm$ 40.8 | **97.8** $\pm$ 5.0 |
| Drawer Open | 20K | 99.9 $\pm$ 0.3 | 65.3 $\pm$ 41.6 | 72.0 $\pm$ 45.4 | **89.9** $\pm$ 31.6 |
| Sweep Into | 10K | 88.8 $\pm$ 29.9 | 68.4 $\pm$ 35.2 | 55.7 $\pm$ 48.1 | **88.0** $\pm$ 31.0 |
| Window Open | 1K | 99.9 $\pm$ 0.3 | 55.9 $\pm$ 45.1 | 66.1 $\pm$ 36.5 | **70.7** $\pm$ 40.8 |
| Hammer | 10K | 91.5 $\pm$ 26.5 | 31.0 $\pm$ 24.7 | 30.8 $\pm$ 35.4 | **48.8** $\pm$ 41.2 |

**Table 3: Success rates in Meta-World manipulation tasks.** Two elements in each cell denote the average value and standard deviation of success rates across runs with 10 random seeds.

method in the real-world using a UR-5 robot for a block placing task. Experiment details and additional analyses with rendered results are provided in the supplementary material.

## 5.1 Experiment Settings

**Tasks.** Locomotion tasks from DMControl consist of walker walk, cheetah run, quadruped walk, humanoid walk, and hopper hop. Manipulation tasks from Meta-World consist of button press, door open, drawer open, sweep into, window open, and hammer. For each task, we train with 10 different seeds used in prior work [6, 10] and measure the average performance with a standard deviation. Evaluation metrics are chosen as the ground-truth return per episode for DMControl and the success rate for Meta-World. Additionally, we conduct experiments with real human feedback for 30 participants using policy and reward models pre-trained in the simulation environment. We also demonstrate a block placing task using a UR-5 robot in the real-world, by pre-training the policy in the Mujoco simulator [27] and fine-tuning in the real-world. Detailed experiment settings are provided in the supplementary material.

**Baselines and Implementation Details.** We use three baselines for all tasks: SAC [28], Meta-Reward-Net (MRN) [10], and PEBBLE [7]. Agents learned via SAC with the true reward functions are considered as the upper bound of all tasks. We use MRN and PEBBLE as the baseline methods using the traditional trajectory comparison method, pairwise comparison. MRN utilizes bi-level optimization for both reward and policy learning and achieves state-of-the-art among previous methods. We apply our sequential preference ranking algorithm on both MRN and PEBBLE to show the effectiveness of our algorithm. Results with MRN baseline are described in this section and results with PEBBLE baseline are provided in the supplementary material. We use unsupervised pre-training [7] for 9,000 steps for all experiments. The trajectory encoder is implemented using a two-layer feed-forward neural network, where the input dimension is the combined size of the state and action spaces, and the output dimension is set to 256. The linear reward model then takes the encoded feature and passes it through a single fully connected neural network. For each task, hyperparameters and implementation details are described in the supplementary material.

## 5.2 Simulation Experiments

**Locomotion Tasks from DMControl.** Results in Table 2 demonstrate the superiority of the proposed root pairwise comparison over the baseline pairwise comparison and sequential pairwise comparison in all locomotion tasks. The reward obtained after convergence is higher by an average of $29.0\%$ compared to conventional pairwise comparison used in MRN [10]. Sequential pairwise comparison, on the other hand, does not exceed root pairwise comparison, but it improves performance against pairwise comparison by $10.3\%$ on average. Based on the standard deviation of rewards after convergence across 10 different runs, root pairwise comparison shows higher stability than sequential

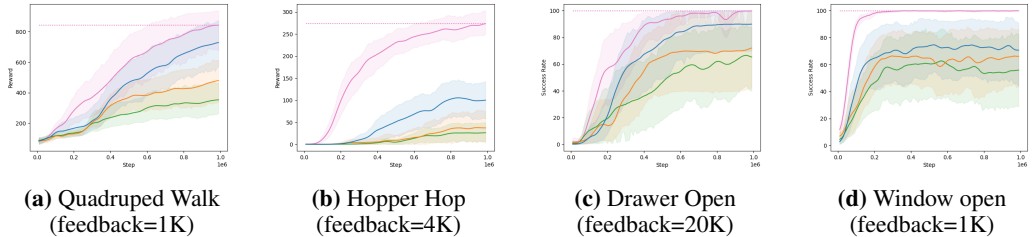

**(a)** Quadruped Walk
(feedback=1K)

**(b)** Hopper Hop
(feedback=4K)

**(c)** Drawer Open
(feedback=20K)

**(d)** Window open
(feedback=1K)

**Figure 3: Comparison results in two locomotion tasks from DMControl and two manipulation tasks from Meta-World.** All methods are implemented based on MRN. Each subfigure describes the comparison results among three trajectory comparison methods: pairwise (green), sequential pairwise (orange), and root pairwise (blue). Pink lines describe the oracle performance using SAC with the true reward.

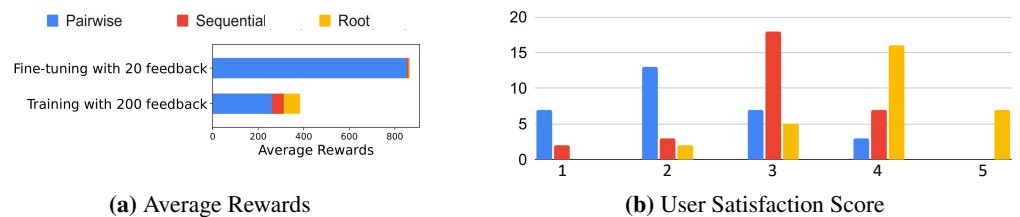

**(a)** Average Rewards

**(b)** User Satisfaction Score

**Figure 5: Training the Cheetah Run Task with Real Human Feedback.** (a) The bar plot describes the ratio between the average rewards after training the policy with real human feedback. (b) The histogram describes distributions of user satisfaction scores for three trajectory comparison methods.

pairwise comparison. Reward graphs in Figure 3 (a) and (b) show that root pairwise comparison enables faster convergence of the policy learning.

Additionally, we assess the converged reward for various feedback maximums (0.1K, 1K, 2.5K, 4K) in the quadruped walk task, as illustrated in Figure 4. The graph reveals that both sequential and root pairwise comparison methods converge to higher average reward values and lower variance as the number of feedbacks increases. This implies that with a sufficient number of feedbacks, the amount of augmented preference data affects performance more than data dependency does. By taking advantage of learning a dense reward [5, 29, 30], root pairwise comparison outperforms the oracle baseline with a sufficient number of feedbacks.

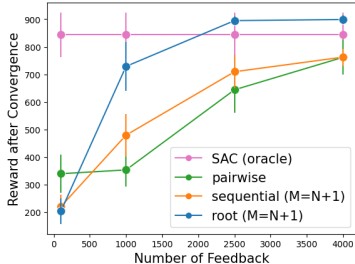

**Figure 4: Average reward per number of feedbacks after convergence.**

**Manipulation Tasks from Meta-World.** Results in Table 3 show the comparison results with baseline in manipulation tasks. The proposed method using root pairwise comparison outperforms the methods using pairwise comparison and sequential pairwise comparison in all tasks, improving the overall success rate by $25.0\%$ compared to the pairwise comparison baseline. Reward graphs in Figure 3 (c) and (d) show that root pairwise comparison enables faster convergence of the policy. Unlike the locomotion tasks, sequential pairwise comparison only improves the success rate by $1.2\%$ over pairwise comparison. This may be due to the generalization error using sequential pairwise comparison. Comparing the standard deviation of success rates among 10 different runs, root pairwise comparison achieves the lowest standard deviation. This implies that root pairwise comparison is the most stable algorithm for manipulation tasks among three trajectory comparison methods.

**Experiments with Real Human Feedback.** To compare the performance and the user stress levels associated with three different methods: pairwise, sequential pairwise, and root pairwise comparison, we conduct experiments involving 35 real human participants. 5 participants trained the policy and the reward models from scratch with 200 human feeedbacks. Other participants fine-tuned the models with 20 human feedbacks, where the models were pre-trained with simulated feedbacks. Figure 5 (a) shows that the proposed methods outperform the baseline in both training from scratch and fine-tuning experiments. Especially for training the models from scratch, root and sequential

| Method | # feedback | Success Rate↑ | True Reward↑ | Episode Length↓ | Reward Accuracy↑ |
|---|---|---|---|---|---|
| Pairwise | 0.8K | 34.4 | 17.1 | 29.6 | 75.8 |
| Sequential Pairwise | 0.8K | 41.0 | 20.1 | 28.4 | 80.9 |
| Root Pairwise | 0.8K | **50.5** | **25.9** | 27.9 | **84.9** |

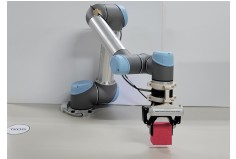

**Table 4: Average performance in the real robot manipulation task.** For each method, the agent is fine-tuned in the real world for 3,000 steps.

**Figure 6: Block placing using a UR-5 robot.**

pairwise comparison outperform conventional pairwise comparison by $47.7\%$ and $38.6\%$, respectively. Quantitative results are provided and analyzed in the supplementary material. Additionally, as a user study, we asked the participants to rate their user satisfaction on a scale from one to five, with higher scores indicating greater satisfaction and lower stress levels. Figure 5 (b) illustrates that the average user satisfaction scores resulted in 2.20, 3.00, and 3.93 for pairwise, sequential, and root pairwise comparison, respectively. We perform paired-sample t-test and the differences between the scores are statistically significant for all cases, with the average p-value 0.035. Specifically, users responded that it is convenient to compare with one's most preferred trajectory. These results imply that root pairwise comparison is the least burdensome for real human users while achieving the highest performance.

### 5.3 Real Robot Experiments

**Block Placing Task.** To demonstrate our method in real-world environments, we conduct a block placing task using a real UR-5 robot. As illustrated in Figure 6, the goal of this task is to place a block at a given target position starting from a randomly chosen initial position. For faster training and stable movements of the agent, we train the policy with a deep Q-learning network (DQN) and discrete action space following prior work [31]. In this setting, an action is given by the moving direction $\psi$, where $\psi$ is the heading direction of the end effector discretized into eight angle bins $\{0, \frac{\pi}{4}, \frac{2\pi}{4}, ..., \frac{7\pi}{4}\}$. Once $\psi$ is determined, the agent changes its pose and moves the end effector for a fixed distance in the angle of $\psi$. The policy is pre-trained in the simulator for 20,000 steps and fine-tuned in the real-world for 3,000 steps. The reward model is initialized before fine-tuning. Results in Table 4 show the comparison results with baseline after fine-tuning in the real-world. Using root pairwise comparison, the agent exhibits the highest success rate and the reward model achieves the fastest convergence with the highest accuracy among three trajectory comparison methods.

## 6 Conclusion

We propose a novel RLHF framework called SeqRank that utilizes sequential preference ranking, resulting in a substantial improvement in the feedback efficiency by at least $39.2\%$. Both sequential and root pairwise comparison enables faster estimation of the reward function. We also prove the convergence rates of the empirical risk and the generalization bound of the reward model for each method. Theoretical and experimental results show that both methods outperform conventional pairwise comparison, with root pairwise comparison showing the most substantial improvement against the baseline. The proposed method and theoretical analyses can be applied to other domains and serve as a practical solution to enhance the feedback efficiency and performance of RLHF. Potential negative societal impacts include amplifying existing biases present in human preferences. We hope our method encourages RLHF to be used for social good without violating ethics.

**Limitations.** As we experimentally show that the proposed root pairwise comparison achieves the highest reward accuracy among three trajectory comparison methods, we expect to find a tighter generalization bound of the reward model. As the generalization bound of logistic regression with dependent data is not fully studied yet, it will be our future work.

## Acknowledgments and Disclosure of Funding

This work was in part supported by Institute of Information & Communications Technology Planning & Evaluation (IITP) grant funded by the Korea government (MSIT) (No. 2019-0-01190, [SW Star Lab] Robot Learning: Efficient, Safe, and Socially-Acceptable Machine Learning, 66%, No. 2022-0-00331, Development of emotion recognition/generation-based interacting edge device technology for mental health care, 17%, and No. 2021-0-01341, AI Graduate School Program, CAU, 17%).

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
