# OpenReview forum: "Sequential Preference Ranking for Efficient Reinforcement Learning from Human Feedback"
_NeurIPS.cc/2023/Conference — NeurIPS 2023 poster_

### Official Review · Reviewer_f3sA · 2023-07-05

**Soundness:** 3 good
**Presentation:** 3 good
**Contribution:** 3 good
**Rating:** 6
**Confidence:** 3

**Summary:**

The authors propose that a novel RLHF framework called SeqRank to improve the feedback efficiency via sequential preference ranking. They propose two methods: sequential pairwise comparison and root pairwise comparison.

**Strengths:**

The paper is clear and writing is easy to follow. The main idea is theoretically sound and the experiment results are promising.

**Weaknesses:**

The proposed method seems to be relatively simple. But I think it’s okay since the authors prove its effectiveness both theoretically and empirically.

**Questions:**

1 the proposed methods (sequential pairwise comparison, root pairwise comparison) is quite intuitive. Though it achieves better performance than standard pairwise comparison, it might be suboptimal.

2 the definition of $S_M$ is not clear (number of pairs collected from $M$ trajectories). Here “pairs” refers to both directly compared pairs and indirectly inferred pairs (e.g., the number of lines in both black and purple in fig2(b)), is that correct?

3 As for experiments, the numbers of feedbacks different from tasks. How do you choose this parameter? In figure 4, when the number of feedbacks is limited, it’s interesting to see the pairwise comparison outperform the two sequential methods. Is there any comment on this?

4 As for the writing, since the proposed method aims to improve the feedback efficiency, which seems to be defined by the authors, I suggest the author provide some illustrations first to show its importance (e.g., higher efficiency will lead to higher performance) before discussing the proposed method is able to improve it. This might make it more clear.

---

> ### Author Rebuttal · Authors · 2023-08-09
>
> We truly appreciate the positive and thoughtful review by Reviewer f3sA, _e.g., clear and easy writing, theoretically sound idea, and promising experiment results_. We will address all raised concerns by the reviewer and revise the paper accordingly.
>
> ## **Weakness**
> **Weak1: The proposed method seems to be relatively simple. But I think it’s okay since the authors prove its effectiveness both theoretically and empirically.**
> > We are pleased to Reviewer f3sA’s comments that the reviewer recognizes the theoretical soundness of the proposed method, and acknowledges the theoretical and empirical effectiveness of our method. Our intention is indeed to develop a method that is not only simple but also effective. We believe that this simplicity can be a strength, facilitating implementation and adaptability across different domains, while the robust theoretical and empirical results ensure that the proposed method significantly improves the performance in DMControl, Meta-World, and real-world robotic manipulation.
>
> ## **Questions**
> **Question1: the proposed methods (sequential pairwise comparison, root pairwise comparison) is quite intuitive. Though it achieves better performance than standard pairwise comparison, it might be suboptimal.**
>
> > Yes, our work can be seen as suboptimal because it is a fundamental research about augmenting preference ranking data without additional human cost. However, we expect various applications of our work by combining the proposed sequential preference ranking with existing sampling methods, i.e., uncertainty-based disagreement sampling. To support our claim, we perform additional experiments by applying disagreement sampling to our method in **Global response to All reviewers - Additional Experiments - Exp2**. Additionally, we can improve the algorithm by using sequential/root pairwise comparison to augment preference data until the data dependency exceeds a threshold value. After exceeding the threshold value, we can switch the trajectory comparison method to pairwise comparison. By scheduling between two or more trajectory comparison methods, we will be able to develop an RLHF algorithm that effectively augments preference data with limited data dependencies.
>
> **Question2: the definition of $S_M$ is not clear (number of pairs collected from trajectories). Here “pairs” refers to both directly compared pairs and indirectly inferred pairs (e.g., the number of lines in both black and purple in fig2(b)), is that correct?**
> > Yes, the definition of $S_M$ includes both the number of directly compared pairs and indirectly compared pairs. We will revise the definition of $S_M$ described in L163 in the paper and Table 1 in the supplementary material as follows:\
> \
> $S_M$: the total number of segment pairs from $M$ trajectory segments including both directly and indirectly compared pairs.
>
> **Question3: As for experiments, the numbers of feedbacks different from tasks. How do you choose this parameter? In figure 4, when the number of feedbacks is limited, it’s interesting to see the pairwise comparison outperform the two sequential methods. Is there any comment on this?**
>
> > For the locomotion tasks in DMControl, we perform breadth-first search (BFS) for the number of feedbacks in the range of [200, 40000]. For the manipulation tasks in Meta-World, we perform BFS for the number of feedbacks in the range of [1000, 20000]. For agents with simple dynamics or tasks, we choose a low number in the range and for agents with complex dynamics or tasks, we choose a high number in the range.\
> \
> For the experiments with real human feedback, we set the number of feedbacks as 20 for fine-tuning the policy and the reward model. For the real robot experiments, we perform BFS for the number of feedbacks in the range of [200, 800].\
> \
> In Figure 4, we fix the frequency of reward learning as learning the reward at each 30,000 policy training steps for fair comparison among different numbers of feedback. For a small number of feedback, the increased data dependency affects the performance more than the increased feedback efficiency. In practical cases, we can learn the reward function more frequently to improve performance if the maximum number of feedback is as low as 100. Please refer to **Global Response to All Reviewers (Part I)-B.1**.
>
> **Question4: As for the writing, since the proposed method aims to improve the feedback efficiency, which seems to be defined by the authors, I suggest the author provide some illustrations first to show its importance (e.g., higher efficiency will lead to higher performance) before discussing the proposed method is able to improve it. This might make it more clear.**
>
> >Yes, we will add illustrations in Figure 1 in the main paper to describe that higher efficiency will lead to higher performance. This will definitely improve the understanding about the importance of feedback efficiency in reinforcement learning from human feedback (RLHF).

---

> > ### Comment · Reviewer_f3sA · 2023-08-21
> > **Response to the rebuttal**
> >
> > Thanks to the authors for the detailed response. I appreciate the additional clarifications from the authors.

---

> > > ### Author Response · Authors · 2023-08-21
> > > **Thank you for your response**
> > >
> > > We are grateful for Reviewer f3sA's appreciative feedback and recognition of the clarifications provided in our rebuttal. The reviewer's constructive insights and feedback significantly improved the quality of our work.

---

> ### Comment · Area_Chair_Quzi · 2023-08-21
>
> Dear Reviewer,
>
> Thanks for your effort in reviewing. Could you take a look at the authors' response and check whether your concerns were addressed?
>
> Thanks!

---

### Official Review · Reviewer_Mtqb · 2023-07-05

**Soundness:** 3 good
**Presentation:** 3 good
**Contribution:** 2 fair
**Rating:** 7
**Confidence:** 4

**Summary:**

This paper introduces a novel approach, named SeqRank, to enhance the efficiency of reinforcement learning from human feedback. By incorporating sequential preference ranking, the proposed method significantly improves average feedback efficiency.  Evaluations demonstrate its superiority over existing baselines in simulated locomotion and manipulation tasks, as well as in real-world scenario with a robot manipulation task.

**Strengths:**

The paper exhibits strengths in several areas. Firstly, it effectively identifies and defines relevant problem o feedback efficiency, Secondly, the paper provides a strong mathematical justification for the proposed method. The authors present a theoretical framework, underpinned by sound mathematical principles. Furthermore, the paper showcases evaluations conducted on a diverse set of tasks, including real-world experiments with a physical robot. The inclusion of real-world experiments adds a practical element, validating the effectiveness of the proposed solution in realistic settings.

**Weaknesses:**

One limitation of the paper is the lack of consideration for alternative sampling methods beyond pairwise sampling. While uniform pairwise sampling is a widely used, other sampling methods, such as Uncertainty-based sampling[1] perform better.

Another point that could have been addressed in the paper is the possibility to label comparisons as "equal”.  It is worth mentioning that incorporating this labeling aspect could potentially lead to improvements in performance of the proposed method, but it is not clear how that could be done.

[1] Kimin Lee, Laura Smith, Anca Dragan, and Pieter Abbeel. B-pref: Benchmarking preference-based reinforcement learning. Neural Information Processing Systems Track on Datasets and Benchmarks(NeurIPS), 2021

**Questions:**

1. In Section 5.2, you mentioned that Sequential pairwise comparison performs better than pairwise comparison on average. However, in several tasks, such as Sweep Into, pairwise comparison was better on average than sequential pairwise. Could the authors please explain why this discrepancy exists?

2. In Figure 4, authors presented results for a specific task. Based on the values shown, it appears to be the Quadruped Walk task. However, it is not explicitly mentioned in the paper which task the results in Figure 4 correspond to. Could the authors please clarify this?

3. Regarding the Experiments with Real Human Feedback: what are the standard deviations of the results. Could you provide information on the standard deviation for the reported data? Additionally, have the authors conducted any statistical hypothesis tests to validate the claims made based on the experiments with real human feedback?

**Limitations:**

See Weaknesses.

---

> ### Author Rebuttal · Authors · 2023-08-09
>
> We truly appreciate the positive and constructive review by Reviewer Mtqb, _e.g., strong mathematical justification, validation of the effectiveness of the proposed solution in realistic settings_. We will address all raised concerns by the reviewer and revise the paper accordingly.
>
> ## **Weaknesses**
> **Weak1: One limitation of the paper is the lack of consideration for alternative sampling methods beyond pairwise sampling. While uniform pairwise sampling is widely used, other sampling methods, such as Uncertainty-based sampling[1] perform better.**
>
> > As Reviewer Mtqb mentioned, the uncertainty sampling method is used in prior work PEBBLE [1]. We conduct additional experiments by applying uncertainty-based sampling to our method. Please refer to **Global response to All reviewers - Additional Experiments - Exp2**.
>
> **Weak2: Another point that could have been addressed in the paper is the possibility to label comparisons as "equal”. It is worth mentioning that incorporating this labeling aspect could potentially lead to improvements in performance of the proposed method, but it is not clear how that could be done.**
>
> > In B-Pref [2], two trajectory segments are preferred equal if the difference between the rewards corresponding to the two trajectories is less than a threshold value.
> Suppose we sample 7 trajectory segments with the corresponding rewards [4.9, 5, 4, 3, 3.1, 7, 9]  for sequential pairwise comparison. If we apply our original sequential pairwise comparison, we can augment 4 indirect trajectory pairs. However, if we set the equal preference threshold as 0.2 and perform sequential pairwise comparison with additional transitivity assumptions (1) if A=B and B<C, A<C and (2) if A=B and B>C, A>C, we can augment 9 indirect trajectory pairs, which is more than twice as the original number of augmented trajectory pairs. Although our paper mainly focuses on basic settings without considering equal labels, we expect that our method can be combined with various types of simulated teachers suggested in PEBBLE.
>
> ## **Questions**
> **Question1: In Section 5.2, you mentioned that Sequential pairwise comparison performs better than pairwise comparison on average. However, in several tasks, such as Sweep Into, pairwise comparison was better on average than sequential pairwise. Could the authors please explain why this discrepancy exists?**
>
> > As Reviewer Mtqb mentioned, sequential pairwise comparison outperforms pairwise comparison in all DMControl locomotion tasks while underperforms for door open, sweep into, and hammer tasks in Meta-World. For door open and hammer tasks, the differences between the average success rates are only 0.5 and 0.2, respectively, which could be seen as noise. Still, for sweep into task, the discrepancy gets higher because the true reward function from the simulation environment is not as good as other tasks. Based on the performance of the Oracle SAC model with true reward function in sweep into task, the success rate is 88.8%, which is the lowest among all manipulation tasks in Meta-World. With an improved true reward function that can represent human expertise, we expect that the discrepancy between pairwise and sequential pairwise comparison will be reduced, or sequential pairwise comparison will perform better than conventional pairwise comparison.
>
> **Question2: In Figure 4, authors presented results for a specific task. Based on the values shown, it appears to be the Quadruped Walk task. However, it is not explicitly mentioned in the paper which task the results in Figure 4 correspond to. Could the authors please clarify this?**
> > Yes, we apologize for the lack of information about the task. Figure 4 illustrates the experiment results in DMControl quadruped walk task with various number of feedbacks. Please refer to Global response to All Reviewers - B. Additional Experiments and Analyses - Additional Analysis for Figure 4 in the main paper.
>
> **Question3: Regarding the Experiments with Real Human Feedback: what are the standard deviations of the results. Could you provide information on the standard deviation for the reported data? Additionally, have the authors conducted any statistical hypothesis tests to validate the claims made based on the experiments with real human feedback?**
>
> > We include statistical analysis for the real human experiment. Please refer to **Global Response to All Reviewers (Part II)-B.2**. Also, we trained the agent with real human feedback from scratch during the rebuttal period, without using any pre-trained policy or reward models. Please refer to **Global Response to All reviewers - Additional Experiments - Exp3**.
>
> [1] Lee et al., "PEBBLE: Feedback-Efficient Interactive Reinforcement Learning via Relabeling Experience and Unsupervised Pre-training", International Conference on Machine Learning, 2021.\
> [2] Lee et al. "B-pref: Benchmarking preference-based reinforcement learning", Neural Information Processing Systems Track on Datasets and Benchmarks(NeurIPS), 2021.

---

> > ### Comment · Reviewer_Mtqb · 2023-08-16
> >
> > Thanks to the authors for the detailed answers that successfully clarified the issues that were originally raised. Therefore, I would increase my rating to Accept.

---

> > > ### Author Response · Authors · 2023-08-17
> > > **Thank you for your response**
> > >
> > > We thank Reviewer Mtqb's kind review and acknowledgment of the responses in our rebuttal, and we sincerely appreciate the reviewer’s willingness to raise the rating based on our response. The reviewer's constructive feedback greatly enhanced the quality of our work.

---

### Official Review · Reviewer_tocg · 2023-07-06

**Soundness:** 3 good
**Presentation:** 3 good
**Contribution:** 3 good
**Rating:** 6
**Confidence:** 3

**Summary:**

This paper studies the idea of the feedback efficiency of RLHF, which the authors define as the ratio between the number of trajectory pairs divided by the number of feedbacks. To optimize such a metric, the authors propose new algorithms that decide which trajectories to compare. In particular, the authors propose a sequential pairwise comparison strategy, where the most recent trajectories are compared and a root comparison strategy, where the defender is the previously most preferred trajectory.


**Strengths:**

- This paper studies the important and timely problem of RLHF.
- The paper has extensive experiments, including simulated and real world tasks, as well as using real human feedback.
- The study on the labelers satisfaction is novel and very interesting.

**Weaknesses:**

**W1.** One major concern with this paper is that the claim of completeness, (that any two trajectories are comparable) is incorrect for the main RLHF application of LLM finetuning [1]. Since this is the claim is key to the paper, I think it makes the paper poorly motivated/placed in the literature.

[1] Ouyang, Long, et al. "Training language models to follow instructions with human feedback." Advances in Neural Information Processing Systems 35 (2022): 27730-27744.
APA

**W2.** Another major issue is that the contribution of using the transitive property of human preferences is limited. This is a limited contribution since RLHF works typically design the labeling interface in such a way that the labeler will implicitly use the transitive property of human preferences. An example can be seen in Figure 12 of the InstructGPT paper [1].

**W3,** The claim that cognitive overload is a critical drawback of RLHF seems like an overstatement. In fact, the labeler satisfaction and inter-labeler agreement in the InstructGPT paper (and other RLHF papers) is high (see Table 13 of [1]).

**W4.** The presentation of the author's method is not very clear and the theoretical results are hard to follow. The main tool the authors use to illustrate their method is Figure 2, which is not exact enough. Moving the algorithm to the main text would be helpful.



**Questions:**

**Q1.** What would be the impact of these algorithms in settings where only a small number of trajectories are comparable?

**Q2.** Can you show RM convergence is faster (similar to figure 4)? The algorithms proposed in this paper should most directly affect the RM convergence.

**Q3.** In figure 2, it seems like pairwise comparison would result in a more diverse training set. Could this diversity boost the performance of the reward model?

**Limitations:**

Yes, the authors address the limitations of the study.

---

> ### Author Rebuttal · Authors · 2023-08-09
>
> We truly appreciate all feedback from Reviewer tocg and will incorporate the reviewer’s comments to revise our paper to be more sound and clear. Since the reviewer’s concerns are mainly grounded in the InstructGPT paper [1], we would like to mention three major differences between the robotics domain and the language domain RLHF.
>
> **1. Data and Feedback Complexity:**
> Gathering data and feedback in robotics requires the assessment of complex physical interaction with the environment, whereas LLM relies on generative models.\
> **2. Transitivity Property and Ranking:**
> Applying transitivity property via ranking feedback is more complex in robotics due to sequential actions, compared to the more straightforward application in LLM.\
> **3. Generalization:**
> Generalizing across tasks is more complex in robotics due to specific physical properties, whereas it is more feasible across tasks in LLM. Therefore, in the robotics domain RLHF, we deal with a specific fundamental task using a robot with fixed dynamics in a fixed environment.
>
> ## **Weaknesses**
> **Weak1: One major concern with this paper is that the claim of completeness … incorrect for the main RLHF application of LLM finetuning [1]. … it makes the paper poorly motivated/placed in the literature.**
>
> >Based on the InstructGPT paper [1], RLHF is originally developed for training simple robots and Atari games, and has recently diverged into two parts: (1) its original purpose and (2) fine-tuning language models. Our primary focus is on training an agent for specific robotic tasks, like drawer opening with a robot arm (manipulator) with fixed dynamics in a fixed environment. In such contexts, preferences are understood as judgments on robot performance. Based on our knowledge, all previous RLHF work in robotics (a.k.a. PBRL) assume that any two trajectories are comparable in a specific task. Therefore, we consider the completeness assumption reasonable and the transitivity property of preference is applicable in our work.
>
> **Weak2: Another major issue is that the contribution of using the transitive property of human preferences is limited. This is a limited contribution since RLHF works typically design the labeling interface in such a way that the labeler will implicitly use the transitive property of human preferences. An example can be seen in Figure 12 of the InstructGPT paper [1].**
>
> >As Reviewer tocg mentioned, transitivity is a natural property of human preference, and humans implicitly use transitivity when they provide ranking feedback. While recent work [1] in the language domain RLHF uses ranking feedback, there isn’t a successful application of the ranking feedback in robotics yet due to the three differences between robotics and LLM listed at the top of this thread. Even if a human can provide ranking feedback, pairwise comparison requires less effort than ranking 3+ trajectories. Also, the proposed sequential preference ranking is more robust to noise than ranking feedback. Our methods are less affected by human uncertainty because the agent ranks preferences autonomously, not relying on human’s ranking feedback (L097).
> \
> We claim that Reviewer tocg’s concern that the transitivity property is already used in the LLM domain _emphasizes the motivation and importance of our work_ rather than limits our contribution, especially given the challenges of using ranking feedback in the robotics domain.
>
> **Weak3: The claim that cognitive overload is a critical drawback of RLHF seems like an overstatement. In fact, the labeler satisfaction and inter-labeler agreement in the InstructGPT paper (and other RLHF papers) is high (see Table 13 of [1]).**
>
> >Yes, we will revise the statement. Please refer to **Global Response to All Reviewers (Part I) -A.2**. However, the labeler satisfaction survey in InstructGPT is a domain-specific survey to rank language instructions, not robot trajectories. Considering the difference between the two domains, the survey does not imply that participants involved in robotics domain RLHF experiments could also be highly satisfied using ranking feedback. Also, while pairwise comparison is simpler than ranking feedback to users, we show that our method improves user satisfaction over conventional pairwise comparison.
>
> **Weak4: The presentation of the author's method is not very clear ... Figure 2 is not exact enough. Moving the algorithm to the main text would be helpful.**
>
> >Yes, we will move the algorithm to the main text for a clear description of the proposed method and revise Figure 2 to include how the reward model is trained from the trajectory pairs. Please refer to **Global Response to All Reviewers (Part I) -A.1**.
>
> ## **Questions**
> **Question1: What would be the impact of these algorithms in settings where only a small number of trajectories are comparable?**
> >If two trajectories are too similar in many queries, we can apply "equal" labels and improve our algorithm for data augmentation. Please refer to the response to Reviewer Mtqb-Weak2.
>
> **Question2: Can you show RM convergence is faster (similar to figure 4)? The algorithms ... should most directly affect the RM convergence.**
>
> >Yes, as derived in **Convergence Rate of the Empirical Risk** (L242) in Section 4.2, one of our main contributions is that the reward model converges faster than conventional pairwise comparison. Specifically, root pairwise comparison achieves twice as fast, and sequential pairwise comparison achieves a 39.2% faster convergence rate of the reward model on average.
>
> **Question3: In figure 2, it seems like pairwise comparison would result in a more diverse training set. Could this diversity boost the performance of the reward model?**
> >Diversity depends on both (1) data dependency and (2) the number of data samples. Please refer to **Global Response to All Reviewers (Part V) - Exp4**.
>
> [1] Ouyang et al., "Training language models to follow instructions with human feedback", Advances in Neural Information Processing Systems, 2022.

---

> > ### Comment · Reviewer_tocg · 2023-08-13
> > **Thank you for the response.**
> >
> > Thank you for the detailed response! I am not as familiar with the application of RLHF in robotics, but the author's responses have alleviated my concerns about the setting studied in this paper. I will raise my score accordingly. However, I think there have been a few misunderstandings when addressing my questions:
> >
> > **Q1:** Here I am basically asking how would this algorithm perform when applied to the RLHF application of finetuning language models?
> >
> > **Q2:** Are the results you mention here empirical or theoretical? I am curious if you notice faster RM convergence empirically.

---

> > > ### Author Response · Authors · 2023-08-14
> > > **Thank you for the quick response. (Part 1)**
> > >
> > > We thank Reviewer tocg for the follow-up comments, and we sincerely appreciate the reviewer’s willingness to raise the rating based on our response. We are glad to know that we were able to alleviate the reviewer’s concerns about the setting studied in this paper.
> > >
> > > We apologize for any misunderstandings in addressing the reviewer’s questions. The reviewer’s clarifications are helpful to better understand the questions, and we discuss the questions in more detail below:
> > >
> > > **Q1: What would be the impact of these algorithms in settings where only a small number of trajectories are comparable? Here I am basically asking how would this algorithm perform when applied to the RLHF application of finetuning language models?**
> > >
> > > > First and foremost, we thank Reviewer tocg for giving us the opportunity to think about the **broader impacts** of our work. While our work was initially designed for the robotics domain, we recognize the potential to extend our approach to the RLHF application for fine-tuning language models. By applying our method to LLM, we anticipate **speeding up the few-shot fine-tuning for specific tasks**. As we specify the task (e.g., fine-tuning a general LLM to summarize a paper) and the group of users (from worldwide to students in department A in university B), the comparability among trajectories increases. This enhanced comparability supports the completeness and transitivity assumptions. Therefore, if only a small number of trajectories are comparable, we will specify the downstream task and the group of users to apply our method. Then, the core principles of our algorithm: (1) sequential preference ranking for pairwise comparison, (2) faster convergence, and (3) improved user satisfaction could be adapted to fine-tuning task-specific language models as follows.\
> > > \
> > > **Faster Convergence and Better Performance.**
> > > First, we can develop an LLM fine-tuning algorithm that only requires the user to perform pairwise comparison but **converges faster** and achieves **better performance** after convergence. This efficiency could open new possibilities for more personalized and responsive applications of language models. \
> > > \
> > > **Reducing Human Uncertainty.**
> > > Second, our method can **reduce human uncertainty** compared to the traditional ranking-based RLHF approach [1]. While both methods leverage the transitivity property in human preference, our method does not require a human to rank multiple trajectories by oneself. Instead, because we let the agent **automatically rank trajectories**, we can reduce human uncertainty in the ranking procedure.\
> > > \
> > > **Better User Satisfaction.**
> > > Furthermore, our method **improves user satisfaction** over conventional pairwise comparison in robotics, and similar principles might be applied to gathering human judgments on textual data. This approach can make the labeling process more **intuitive** and **user-friendly**, which could be essential for encouraging broader user participation in model fine-tuning.\
> > > \
> > > We also expect a positive societal impact using our method on toxicity reducing tasks by **improving human feedback efficiency**, where the RLHF algorithm plays a vital role in mitigating the creation of harmful content by language models [2]. This is crucial for ensuring the secure usage of language models in real-world applications.\
> > > \
> > > In conclusion, we believe that our work offers valuable insights and techniques that could be fruitfully applied to the rapidly growing field of LLM. By recognizing the shared principles, we expect to adapt our approach to the language domain successfully.
> > >
> > > [1] Ouyang et al., "Training language models to follow instructions with human feedback", Advances in Neural Information Processing Systems, 2022.\
> > > [2] Zhao et al., "A survey of large language models." arXiv preprint, 2023.

---

> > > > ### Author Response · Authors · 2023-08-14
> > > > **Thank you for the quick response. (Part 2)**
> > > >
> > > > **Q2: Are the results you mention here empirical or theoretical? I am curious if you notice faster RM convergence empirically.**
> > > >
> > > > > The results in our previous response came from theoretical analysis. Based on Reviewer tocg’s comment, we measure the preference loss during reward learning as shown in Figure 6. For the experiment, we set the frequency of reward learning as $1/3000$ in quadruped walk task. Because we can’t modify the uploaded pdf during the discussion phase, we illustrate the figure using characters. The training graphs in Figure 6 show that the reward model converges faster in the order of root pairwise, sequential pairwise, and pairwise comparison. We will include Figure 6 as a real image in the final version of the supplementary material.
> > > >
> > > > ```
> > > > 0         1         2         3         4         5         6         7
> > > > |—-----------------------------------------------------------------------> reward training loss
> > > > |                                  r                          p       s
> > > > |                            r                        p     s
> > > > |                    r                     p s
> > > > |            r                 sp
> > > > |       r            s  p
> > > > |     r       s    p
> > > > |    r     s     p
> > > > |   r    s     p
> > > > |   r  s      p
> > > > |  r  s     p
> > > > |  r s     p
> > > > |  r s   p
> > > > |  r s  p
> > > > |  r s p
> > > > |  r sp
> > > > |  rsp
> > > > |  rsp
> > > > |  rsp
> > > > |  rsp
> > > > | rs p
> > > > | rsp
> > > > | rsp
> > > > | rsp
> > > > | rsp
> > > > | rsp
> > > > |
> > > > v
> > > > policy training step
> > > > ```
> > > > **Figure 6. Reward Training Loss.** x-axis and y-axis describe the reward training loss (0.1 per column) and the policy training steps (3000 steps per row), respectively. ‘r’, ‘s’, and ‘p’ denote root pairwise, sequential pairwise, and pairwise comparison, respectively.

---

### Official Review · Reviewer_sxrB · 2023-07-06

**Soundness:** 3 good
**Presentation:** 2 fair
**Contribution:** 3 good
**Rating:** 7
**Confidence:** 3

**Summary:**

This paper presents SeqRank a framework for labelling user preferences in a ranked fashion taking advantage of the transitive property of human preferences.

Authors mathematically prove that their two variants of SeqRank (sequential pairwise comparison, and root pairwise comparison), achieve better feedback efficiency (defined as the ratio between the number of preference labels and the number of trajectories) than conventional pairwise comparisons. Additionally, authors prove that both methods converge to the same level of empirical risk and that root pairwise comparison converges at the same rate as the baseline pairwise comparison, whereas sequential pairwise converges slightly more slowly.

Empirical evaluations show that both variants of SeqRank beat the performance of a pairwise comparison using a recent Preference-based Reinforcement Learning algorithm  (Meta-Reward Net, MRN), in 5 Mujoco tasks and 6 MetaWorld tasks. Experiments show a correlation between feedback efficiency and preferences sample efficiency.

Additionally, authors prove the usefulness of SeqRank in a finetuning-from-human-feedack task as well as training an actual UR-5 robot on a block-placing task.

**Strengths:**

* Paper presents an intuitive idea (rank preferences) backed up by robust mathematical analysis of its properties.
* Compelling results in a wide variety of tasks, both in simulated and in real-world environments.
* Feedback efficiency seems to correlate with sample efficiency, thus lowering the barrier of adoption for Preference-based Reinforcement Learning methods.

**Weaknesses:**

* Authors only test SeqReq on Meta-Reward Net. The paper would be more significant if  the same benefits observed for MRN+SeqRank were shown to carry over to other PbRL methods such as PrefPPO, PEBBLE [1].
* In my opinion, the experiments with real humans do not help in establishing the significance of SeqRank.
    * The relative improvement with root sequence comparison is of just 0.4% over the reward of the policy before fine-tuning.
    * No measure of spread for the reward reported, so it is not possible to establish whether that improvement is significant or just noise.
    * Regarding the questionnaires, authors should report whether the differences are statistically significant (preferably through a rank-based statistical test).
* Beyond the above issues, I would have found it more significant to train the policy from scratch with actual human preferences and SeqRank as is common in the literature [1,2,3], rather than fine-tuning an already trained policy.


[1] K Lee, L Smith, P Abbeel (2021) PEBBLE: Feedback-Efficient Interactive Reinforcement Learning via Relabeling Experience and Unsupervised Pre-training. ICML.

[2] PF Christiano, J Leike, T Brown, M Martic, S Legg, D Amodei (2017) Deep Reinforcement Learning from Human Preferences. NeurIPS.

[3] B Ibarz, J Leike, T Pohlen, G Irving, S Legg, D Amodei (2018) Reward learning from human preferences and demonstrations in Atari. NeurIPS


**Update after rebuttal**

The authors carried out additional experiments which addressed most of my concerns. In particular, the authors showed that i) SeqRank is compatible with uncertainty sampling, ii) that PEBBLE+SeqRank improves over vanilla PEBBLE, and  iii) that SeqRank outperforms pairwise comparison when training from scratch from human feedback.

The experiments with real human feedback from a pre-trained policy do not show a significant improvement over the baseline, but do show statistically significant higher user satisfaction scores for SeqRank.


**Questions:**

1. PEBBLE does a selection of trajectory pairs based on the reward disagreement between three modules. How does this fit with SeqRank? Does it affect the uniformness assumption? Or does SeqRank simply work off the selected subset?
2. In figure 4, are the task rewards normalised before being averaged? If not, tasks like Walker Walk (where SeqRank does really well) may be overrepresented.
3. Section 5.3 is really compressed, to the point it is difficult to understand exactly what is the experiment.
    * How are the preferences obtained?
    * Is the reward also fine-tuned when training on the actual robot? What is meant by "the reward model is initialized before fine-tuning"?
    * What is True Reward and Reward Accuracy in Table 4?

**Nitpicks and suggestions:**
* In the introduction, the paper claims that obtaining multiple preferences from a single trajectory reduces the cognitive load for the user. The experiments offer no evidence for this. I would rephrase as users prefer a more efficient method.
* Consider simplifying the notation for the expectation for trajectory pairs which is variously $\mathbb{E}[S_N]$, $p_N$, $a_N$, $b_N$  in different parts of the paper. I understand each is subtly different in each context, but perhaps a superscript could be used to denote the differences?
* The description of the SeqRank algorithm is left in the appendix, is there some way to bring it back to the main text?
* Mention in the main text that statistics about the population are in the appendix.
* In figure 4, why is root pairwise comparison beating SAC?


**Update after rebuttal:**

Refer to author's rebuttal for answers.

**Limitations:**

Limitations are adequately addressed. I would point out that this method lowers the barrier of entry to control complex systems that would otherwise not be controlled by humans. This has safety and ethical implications that should be considered before deployment.

---

> ### Author Rebuttal · Authors · 2023-08-09
>
> We truly appreciate the positive and thoughtful review by Reviewer sxrB, _e.g., robust mathematical analysis and compelling results in a wide variety of tasks_. We will address all raised concerns by the reviewer and revise the paper accordingly.
>
> ## **Weaknesses**
> **Weak1: “Authors only test SeqRank on Meta-Reward Net. The paper would be more significant if the same benefits observed for MRN+SeqRank were shown to carry over to other PbRL methods such as PrefPPO, PEBBLE [1].”**
>
> > We thank Reviewer sxrB for suggesting additional experiments to evaluate the effectiveness of our method in various baselines. Based on the reviewer’s suggestion, we conducted experiments using the PEBBLE [1] baseline. Please refer to **Global Response to All reviewers - Additional Experiments - Exp1**. As shown in Table 1 and Table 2, we were able to improve the performance against PEBBLE. We will include experiment results using both PrefPPO and PEBBLE baselines for the final version of our paper.
>
> **Weak2: In my opinion, the experiments with real humans do not help in establishing the significance of SeqRank. … authors should report whether the differences are statistically significant.**\
> **& Weak3: I would have found it more significant to train the policy from scratch with actual human preferences … rather than fine-tuning an already trained policy.**
>
> > We include statistical analysis for the real human experiment. Also, we trained the agent with real human feedback from scratch during the rebuttal period, without using any pre-trained policy or reward models. Please refer to **Global response to All reviewers - B.2. and Exp3**.
>
> ## **Questions**
> **Question1: PEBBLE does a selection of trajectory pairs based on the reward disagreement between three modules. How does this fit with SeqRank? Does it affect the uniformness assumption? Or does SeqRank simply work off the selected subset?**
>
> >As Reviewer sxrB mentioned, uncertainty-based sampling methods such as disagreement sampling in PEBBLE can definitely be applied to SeqRank. We implement disagreement sampling by randomly sampling a subset of trajectories and choosing a trajectory with the highest disagreement with the current defender as the challenger. Implementation details and the experiment results will be included in the final version of the supplementary material. For the experiment results and analysis, please refer to **Global response to All reviewers - Additional Experiments - Exp2**.\
> \
> On the other hand, disagreement sampling does affect the uniformness assumption, therefore we need some modifications on the assumptions of our theoretical analysis. We can first calculate the expected number of trajectory pairs S_M and data dependency based on the sampling strategy. Then, we can just use the convergence rate of the reward model’s empirical risk (L246) and the generalization bound (L255, Theorem 3) to theoretically demonstrate the effectiveness of SeqRank with disagreement sampling.
> Even for other types of sampling strategies, we can follow the same process and easily compare different types of sampling strategies for our work. We claim that Theorem 3 can be widely used to expand our work.
>
> **Question2: In figure 4, are the task rewards normalised before being averaged? If not, tasks like Walker Walk (where SeqRank does really well) may be overrepresented.**
>
> > Yes, the true task rewards from DMControl simulator are normalized before being averaged. We will include this detail in the experiment details in the supplementary material.
>
> **Question3: Section 5.3 is really compressed, to the point it is difficult to understand exactly what is the experiment. How are the preferences obtained? Is the reward also fine-tuned when training on the actual robot? What is meant by "the reward model is initialized before fine-tuning"? What is True Reward and Reward Accuracy in Table 4?**
>
> > The true reward is a hand-designed reward defined by human. Because it takes a long time to train a robot in the real world from scratch, we pre-trained the policy using RLHF with synthetic feedback from the true reward. Then, we fine-tuned the agent policy from the pre-trained model and trained the reward model from scratch. We chose to learn the reward model from scratch because there are some noises in real-world states and actions. We also used synthetic feedback from the true reward function for real-world experiments, due to time constraints. We changed the reward function used in previous work [2] to fit into our block-placing task. The reward accuracy measures the consensus between ensemble networks for the reward model. We will add the specific definitions of the terminologies in the final version of the supplementary material.
>
> ## **Nitpicks and suggestions**
> >**Cognitive Load Statement**:
> Yes, we will rephrase the statement. Please refer to **Global response to All reviewers - A.2. Overstatement Regarding Cognitive Load**.
>
> >**Notation Simplification**:
> We agree with the reviewer’s suggestion to use a superscript to denote the differences between similar notations. We will change the notations as $S_M^*$, where $*$ will be pair, seq, and root for pairwise, sequential pairwise, and root pairwise comparison, respectively.
>
> >**Moving Algorithm to the Main Text**:
> Yes, we will move the algorithm to the main paper. Please refer to **Global response to All reviewers - A.1. Moving Algorithm to Main Text**.
>
> >**Experiment Details**:
> Yes, we will mention that statistics about the real human experiment participants are provided in the supplementary material.
>
> >**Figure 4 - why is root pairwise comparison beating SAC?**:
> Please refer to **Global response to All reviewers - B.1. Additional Analysis for Figure 4 in the main paper**.
>
> [1] Lee et al., PEBBLE, ICML, 2021. (shortened due to characters limit)\
> [2] Kee et al., "Sdf-based graph convolutional q-networks for rearrangement of multiple objects", International Conference on Robotics and Automation, 2023.

---

> > ### Comment · Reviewer_sxrB · 2023-08-15
> > **Thank you for your detailed rebuttal**
> >
> > I would like to thank the authors for their extensive and detailed rebuttal. The new experiments strengthen the significance of the work. In particular, the authors showed that i) SeqRank is compatible with uncertainty sampling, ii) that PEBBLE+SeqRank improves over vanilla PEBBLE, and iii) that SeqRank outperforms pairwise comparison when training from scratch from human feedback.
> >
> > I also appreciated the attached figure in the global response.
> >
> > Regarding the statistical analysis in **B.2**, have the authors applied the Bonferroni correction for repeated statistical tests over the same data?
> >
> > In any case, I agree with the authors assessment that: "it is hard to check statistically meaningful performance improvement using our method by fine-tuning a pre-trained policy because the improvement would be less than training a model from scratch", but at least the experiments do show statistically significant higher user satisfaction scores for SeqRank.
> >
> > Since authors have addressed most of my concerns, I will increase my rating.

---

> > > ### Author Response · Authors · 2023-08-15
> > > **Thank you for your quick response**
> > >
> > > We thank Reviewer sxrB's thoughtful review and acknowledgment of the additional experiments and analyses in our rebuttal, and we sincerely appreciate the reviewer’s willingness to raise the rating based on our response. The reviewer's constructive feedback for additional experiments and analyses significantly improved the soundness of our work. Our response to the reviewer's additional question is as follows.
> > >
> > > **Additional Question: Regarding the statistical analysis in B.2, have the authors applied the Bonferroni correction for repeated statistical tests over the same data?**
> > > > The statistical analysis in B.2 does not use the Bonferroni correction. The corrected p-values using Bonferroni correction (by multiplying `the number of groups=3` on each p-value) for the performance and user satisfaction are in Table 6 and Table 7, respectively. The results in Table 6 imply that the difference is statistically significant for comparing root pairwise comparison to conventional pairwise comparison (paired-sample t-test, corrected p=0.091 < 0.10) with a higher error rate of 10%. On the other hand, results in Table 7 imply that the differences between user satisfaction scores (average values are pairwise: 2.20, sequential: 3.00, and root: 3.93) were still statistically significant for all cases: pairwise vs sequential (p=2.23e-05), pairwise vs root (p=4.91e-07), and sequential vs root (p=6.86e-05) with error rates less than 5%.
> > >
> > > | **group 1** | **group 2** | **statistic** | **p-value** | **p-value with Bonferroni correction** |
> > > |:-----------:|:-----------:|:--------:|:-----------:|:--------------------------------------:|
> > > |   pairwise  |  sequential |   -1.16  |   0.255  |                0.764                |
> > > |   pairwise  |     root    |   -2.28  |   0.030  |                0.091                |
> > > |  sequential |     root    |   1.55   |    0.131  |                0.394                |
> > >
> > > **Table 6. Paired t-test results using Bonferroni correction on rewards after convergence in real human experiments.**
> > >
> > > | **group 1** | **group 2** | **statistic** | **p-value** | **p-value with Bonferroni correction** |
> > > |:-----------:|:-----------:|:--------:|:-----------:|:--------------------------------------:|
> > > |   pairwise  |  sequential |   -5.44  |   7.43e-06  |                2.23e-05                |
> > > |   pairwise  |     root    |   -6.84  |   1.64e-07  |                4.91e-07                |
> > > |  sequential |     root    |   5.04   |   2.29e-05  |                6.86e-05                |
> > >
> > > **Table 7. Paired t-test results using Bonferroni correction on user satisfaction scores in real human experiments.**

---

> > > > ### Comment · Reviewer_sxrB · 2023-08-22
> > > > **Thanks for the clarified statistical analysis**
> > > >
> > > > Apologies for the delay in responding and thank you for clarifying the statistical analysis. The results are in-line with my expectations: not much of an effect on performance, but significant differences on user satisfaction.
> > > >
> > > > I will therefore keep the revised rating.

---

> ### Comment · Area_Chair_Quzi · 2023-08-21
>
> Dear Reviewer,
>
> Thanks for your effort in reviewing. Could you take a look at the authors' response and check whether your concerns were addressed?
>
> Thanks!

---

### Author Response · Authors · 2023-08-09
**Global Response to All Reviewers (Part V)**

### **Exp3. Training the agent with real human feedback from scratch (sxrB-Weak3)**

|   **Task**  | **# feedback** | **reward learning frequency** |  **Pairwise** | **Sequential Pairwise** | **Root Pairwise** |
|:-----------:|:--------------:|:-----------------------------:|:-------------:|:-----------------------:|:-----------------:|
| Cheetah Run |      0.2K      |             1/1000            | 260.2 ± 143.0 |      312.0 ± 276.5      |   **384.3** ± 124.6   |

**Table 5. Experiments with Real Human Feedback.** Two elements in each cell denote the average value and standard deviation of rewards after convergence across 5 participants.

Based on Reviewer sxrB’s suggestion, we train the agent policy and the reward model from scratch using real human feedback. To fit the experiment time in two hours per participant, we set the reward learning frequency as 1/1000 (reward update at each 1000 policy training steps, this value is 1/20000 in simulation).

The results in Table 5 imply that root and sequential pairwise comparison outperform baseline pairwise comparison by 47.7% and 38.6%, respectively. Reward graphs in Figure 3 in the attached pdf show that root pairwise comparison enables faster convergence of the policy, compared to pairwise and sequential pairwise comparison. Additionally, root pairwise comparison shows the most stable training with the lowest variance among participants. Due to the time constraints, we recruited 5 participants for the experiment. We will recruit more participants (>= 25) and update the experiment results with paired-sample t-test results for the final version of our paper.

### **Exp4. Data dependency and the number of augmented data while training the agent (tocg-Question3)**
We measure the number of augmented trajectory pairs and the graph dependency while training the DMControl Quadruped Walk task with 1000 feedback. We set $M=100$ and the reward learning frequency as $1/30000$ so that we sample 100 trajectories at each 30000 policy training steps to train the reward model. All hyperparameters including $M$ are the same as the ones used for the results in Table 2 on page 8 of the main paper. Figure 4 in the attached pdf illustrates the average graph dependency, maximum graph dependency, number of augmented preference data at the current training step, and the cumulative number of augmented preference data.

**Data Dependency.** Theoretically, based on Lemma 3 and Lemma 4 on page 7 of the main paper, the upper bound of maximum graph dependency for sequential and root pairwise comparison is $10.11$ and $195.97$, respectively, when $N=100$, $M=N+1=101$, and $\delta=0.03$. The experiment results in Figure 4 show that the maximum graph dependency for sequential and root pairwise comparison is $9.86$ and $190.43$, respectively. The experiment results align well with the theoretical results and show that our theoretical analysis can be practically used to model the experimental results.

**Number of Augmented Data.** For the number of augmented data at the current training step, the range of the number of augmented data for sequential and root pairwise comparison is $[39.57, 46.00]$ and $[71.71, 640.29]$, respectively. The values are averaged across 10 random seeds. It is notable that the number of augmented data decreases as the training step increases for both sequential and root pairwise comparison. For root pairwise comparison, the number of augmented data at the first reward learning iteration, $640.29$, is $7$ times larger than the expected number of augmented data, $91.61$. On the other hand, for sequential pairwise comparison, the number of augmented data at the first reward learning iteration, $46.00$, is $1.19$ times larger than the expected number of augmented data, $38.75$. The results imply that both proposed methods, especially root pairwise comparison, perform better than the expected scenario and boost the early-stage reward learning process, while the data dependency does not explode and remains in a limited range.

To address the **diversity** that Reviewer tocg mentioned in Question 3, we claim that diversity is affected by both **data dependency** and **the number of augmented data** (which is proportional to feedback efficiency). Because the proposed sequential and root pairwise comparison  maintain low data dependency while augmenting a lot of preference data, both methods outperform the conventional pairwise comparison baseline in most robotic tasks.

---

### Author Response · Authors · 2023-08-09
**Global Response to All Reviewers (Part IV)**

### **Exp2. SeqRank with uncertainty-based disagreement sampling (sxrB-Question1 and Mtqb-Weak1)**

As mentioned in Question 1 from Reviewer sxrB and Weakness 1 from Reviewer Mtqb, we can apply disagreement sampling to the proposed method. We follow the definition of disagreement sampling in PEBBLE, by sampling queries with high disagreement among the ensemble reward models. Considering the available time and computing resources for rebuttal, we perform Exp2 for a reduced number of tasks (3 DMControl and 3 Meta-World) using 10 random seeds. We compare the results with the main experiment results described in Table 2 and Table 3 of the main paper. To compare with the main experiment results, we use MRN [2] as the baseline.

|    **Task**    | **# feedback** | **# training steps** |   **Oracle**  |  **Pairwise** | **Pairwise + disagree** | **Sequential Pairwise + disagree** | **Root Pairwise + disagree** |
|:--------------:|:--------------:|:--------------------:|:-------------:|:-------------:|:----------------------:|:---------------------------------:|:---------------------------:|
|   Cheetah Run  |       10K      |         0.5M         | 705.9 ± 239.8 | 559.2 ± 229.8 |      663.4 ± 56.3      |           676.7 ± 222.1           |        **729.0** ± 120.8        |
| Quadruped Walk |       1K       |          1M          | 843.3 ± 247.3 | 353.5 ± 183.9 |      459.5 ± 238.7     |           456.4 ± 240.0           |        **795.0** ± 190.5        |
|   Hopper Hop   |       4K       |          1M          |  273.5 ± 47.0 |  26.2 ± 37.0  |       15.0 ± 27.9      |            18.2 ± 31.3            |         **114.4** ± 53.0        |

**Table 3. Disagreement Sampling - Rewards after convergence in DMControl locomotion tasks.** Two elements in each cell denote the average value and standard deviation of rewards across runs with 10 random seeds.

|   **Task**   | **# feedback** |  **Oracle** | **Pairwise** | **Pairwise + disagree** | **Sequential Pairwise + disagree** | **Root Pairwise + disagree** |
|:------------:|:--------------:|:-----------:|:------------:|:----------------------:|:---------------------------------:|:---------------------------:|
| Button Press |       10K      |  99.3 ± 0.9 |  95.6 ± 7.6  |       94.9 ± 9.7       |            **100.0** ± 0.0            |         **100.0** ± 0.0         |
|   Door Open  |       10K      | 100.0 ± 0.0 |  77.9 ± 41.3 |       33.3 ± 46.4      |            51.0 ± 49.9            |         **88.2** ± 25.1         |
|    Hammer    |       10K      | 91.5 ± 26.5 |  31.0 ± 24.7 |       32.3 ± 41.3      |            57.5 ± 50.2            |          **97.5** ± 0.7         |

**Table 4: Disagreement Sampling - Success rates in Meta-World manipulation tasks.** Two elements in each cell denote the average value and standard deviation of success rates across runs with 10 random seeds. All tasks are trained for 1M steps.

**Locomotion Tasks from DMControl.** Results in Table 3 demonstrate that root and sequential pairwise comparison with disagreement sampling outperform pairwise comparison with disagreement sampling by 44.0% and 1.0% on average, respectively. While the use of disagreement sampling improves the performance against uniform sampling by only 21.2% for pairwise comparison, the proposed root and sequential pairwise comparison show better performance improvement against pairwise comparison with uniform sampling by 74.5% and 22.6%, respectively. Especially, root pairwise comparison demonstrates its superiority over the pairwise comparison methods (both uniform and disagreement sampling) for all three tasks. Also, for root pairwise comparison, using disagreement sampling improves performance against uniform sampling by 4.5% on average.

**Manipulation Tasks from Meta-World.** Results in Table 4 show that root and sequential pairwise comparison with disagreement sampling outperform pairwise comparison with disagreement sampling by 42.5% and 4.0% on average, respectively. For pairwise comparison, the use of disagreement sampling lowers the average success rate compared to uniform sampling. On the other hand, for root pairwise comparison, using disagreement sampling improves performance against uniform sampling by 4.5% on average.

Performance graphs in Figure 2 in the attached pdf show that root pairwise comparison enables faster convergence of the policy. Due to the time constraints, we demonstrated that MRN+SeqRank using disagreement sampling outperforms the MRN with disagreement sampling for six tasks. For the final version of our paper, we will train and evaluate the model for all tasks and report the results.

[1] Lee et al., "PEBBLE: Feedback-Efficient Interactive Reinforcement Learning via Relabeling Experience and Unsupervised Pre-training", International Conference on Machine Learning, 2021.\
[2] Liu et al., "Meta-reward-net: Implicitly differentiable reward learning for preference-based reinforcement learning", Neural Information Processing Systems, 2022.

---

### Author Response · Authors · 2023-08-09
**Global Response to All Reviewers (Part III)**

### **B.3. Additional Experiments**
We have conducted 4 additional experiments to further validate our approach and respond to specific questions raised by the reviewers:

**Exp1. SeqRank using PEBBLE baseline (sxrB-Weak1)**: We address concerns about the robustness of our approach. The results show that SeqRank performs well with the PEBBLE baseline.\
**Exp2. SeqRank using MRN baseline with uncertainty-based disagreement sampling (sxrB-Question1, Mtqb-Weak1)**: In response to questions about alternative sampling methods, we demonstrate SeqRank using disagreement sampling.\
**Exp3. Training the agent with real human feedback from scratch (sxrB-Weak3)**: Based on Reviewer sxrB's suggestion, we conduct an intensive experiment to train the agent from scratch using real human feedback. The results show that SeqRank not only holds up in simulations but also in practical, real-world applications.\
**Exp4. Data dependency and the number of augmented data while training the agent (tocg-Question3)**: We measure the data dependency and the number of augmented preference data to validate our theoretical results.

***
### **Exp1. SeqRank using PEBBLE baseline (sxrB-Weak1)**

Considering the available time and computing resources for rebuttal, we perform Exp1 in four tasks in DMControl (walker walk, cheetah run, quadruped walk, and hopper hop) and four tasks in Meta-World (door open, drawer open, sweep into, and window open), using 5 random seeds. The oracle performance is also evaluated from the same 5 random seeds. All tasks are trained for 1M steps from scratch.

|    **Task**    | **# feedback** |   **Oracle**  |  **Pairwise** | **Sequential Pairwise** | **Root Pairwise** |
|:--------------:|:--------------:|:-------------:|:-------------:|:-----------------------:|:-----------------:|
|   Walker Walk  |      0.4K      |  957.3 ± 2.3  | 848.1 ± 170.5 |      864.4 ± 120.8      |    **919.8** ± 31.2   |
|   Cheetah Run  |      0.2K      |  886.9 ± 43.2 | 712.5 ± 117.0 |       730.0 ± 96.9      |    **763.5** ± 73.4   |
| Quadruped Walk |       1K       | 843.3 ± 247.3 | 380.7 ± 283.6 |      380.2 ± 319.6      |   **425.1** ± 327.3   |
|   Hopper Hop   |       4K       |  273.5 ± 47.0 |   8.9 ± 8.6   |        0.1 ± 0.1        |    **104.0** ± 3.5    |

**Table 1.  PEBBLE Baseline - Rewards after convergence in DMControl locomotion tasks.** Two elements in each cell denote the average value and standard deviation of success rates across runs with 5 random seeds.

|   **Task**  | **# feedback** |  **Oracle** | **Pairwise** | **Sequential Pairwise** | **Root Pairwise** |
|:-----------:|:--------------:|:-----------:|:------------:|:-----------------------:|:-----------------:|
|  Door Open  |       10K      | 100.0 ± 0.0 |  99.6 ± 0.9  |        99.8 ± 0.4       |    **100.0** ± 0.0    |
| Drawer Open |       20K      | 99.9 ± 0.3 |  82.2 ± 39.2 |       88.8 ± 25.0       |    **100.0** ± 0.0    |
|  Sweep Into |       10K      |  88.8 ± 29.9 |  27.5 ± 47.8 |       41.2 ± 43.2       |    **71.8** ± 48.0    |
| Window Open |       1K       |  99.9 ± 0.3 |  49.4 ± 43.2 |       72.4 ± 39.0       |     **98.6** ± 3.1    |

**Table 2: PEBBLE Baseline - Success rates in Meta-World manipulation tasks.** Two elements in each cell denote the average value and standard deviation of success rates across runs with 5 random seeds.

**Locomotion Tasks from DMControl.** Results in Table 1 demonstrate that root and sequential pairwise comparison outperform baseline pairwise comparison by 14.0% and 1.3% on average, respectively. Especially, root pairwise comparison demonstrates its superiority over the baseline pairwise comparison for all four tasks: walker walk, cheetah run, quadruped walk, and hopper hop. Based on the standard deviation of rewards after convergence in walker walk and cheetah run tasks, root pairwise comparison shows the highest stability among the three trajectory comparison methods. In the hopper hop task, it is notable that only root pairwise comparison achieves an average reward larger than 100 while training the agent using pairwise or sequential pairwise comparison is unsuccessful.

**Manipulation Tasks from Meta-World.** Results in Table 2 show that root and sequential pairwise comparison outperform baseline pairwise comparison by 43.2% and 16.8% on average, respectively. Sequential and root pairwise comparison shows performance improvement using disagreement sampling in door open, drawer open, and window open tasks compared to the results in Table 3 in the main paper using uniform sampling.

Overall, the experiment results in Table 1 and Table 2 show that PEBBLE+SeqRank outperforms the PEBBLE baseline for eight robotic tasks. Performance graphs in Figure 1 in the attached pdf show that root pairwise comparison enables faster convergence of the policy. For the final version of our paper, we will train and evaluate the model using 10 random seeds for all tasks in DMControl and Meta-World, and report the results.

---

### Author Response · Authors · 2023-08-09
**Global Response to All Reviewers (Part II)**

### **B.2. Statistical Analysis of Real Human Experiments (sxrB, Mtqb)**
We agree with Reviewer sxrB’s opinion that the improvements with root pairwise comparison and sequential pairwise comparison are relatively small over the reward of the policy before fine-tuning. The rewards after fine-tuning using pairwise, sequential pairwise, and root pairwise comparison were 860.10 ± 3.35, 860.93 ± 2.28, and 861.70 ± 2.25. While the improvement of the average reward is relatively small, root and sequential pairwise comparison show lower variance compared to conventional pairwise comparison.

We also perform statistical analysis using paired-sample t-test for (1) rewards after convergence and (2) user satisfaction scores. For both metrics, we use the following hypotheses.

- **Null Hypothesis (H0)**: The mean difference between the methods (two chosen from pairwise, sequential pairwise, and root pairwise comparison) in terms of rewards after convergence (or user satisfaction scores) is equal to zero.
- **Alternative Hypothesis (Ha)**: The mean difference between the methods (two chosen from pairwise, sequential pairwise, and root pairwise comparison, same as H0) in terms of rewards after convergence (or user satisfaction scores) is not equal to zero.

The difference is statistically significant for comparing root pairwise comparison to conventional pairwise comparison (paired-sample t-test, p=0.0302 < 0.05). However, the p-value of paired-sample t-test for comparing sequential pairwise comparison to conventional pairwise comparison was calculated as p=0.255, which does not highly guarantee the statistical significance of the performance improvement of sequential pairwise comparison against pairwise comparison.
Based on the main paper’s real human experiments, we found that it is hard to check statistically meaningful performance improvement using our method by fine-tuning a pre-trained policy because the improvement would be less than training a model from scratch. \
\
On the other hand, the differences between user satisfaction scores (average values are pairwise: 2.20, sequential: 3.00, and root: 3.93) were statistically significant for all cases: pairwise vs sequential (p=7.43e-06), pairwise vs root (p=1.64e-07), and sequential vs root (p=2.29e-05). This implies that both root and sequential pairwise comparison improve user satisfaction compared to the baseline trajectory comparison method, pairwise comparison. Among the proposed methods, root pairwise comparison showed the highest user satisfaction score.
\
We will include all statistical analyses in the final version of our paper.

---

### Author Rebuttal · Authors · 2023-08-09

## **Global Response to All Reviewers (Part I)**

We sincerely appreciate all the reviewers for their thoughtful and constructive comments. We are encouraged that they found our motivation and idea clear, intuitive, and novel (sxrB, Mtqb, f3sA) to solve an important and timely problem in RLHF (tocg). We are glad they found our mathematical analysis on the reward model’s (1) convergence rate of the empirical risk, (2) convergence rate of the generalization bound, and (3) the trade-off between feedback efficiency and data dependency strong, robust, and sound (sxrB, Mtqb, fs3A). All reviewers agreed our method shows significant gains in performance across a wide variety of tasks including 11 different robotic tasks in simulation and block-placing task using a real UR-5 robot (sxrB, tocg, Mtqb, f3sA). We appreciate that they found our experiments with real human feedback and user study novel and very interesting (tocg). We will incorporate all feedback in the final version of our paper.

We address common reviewer comments as a **global response** and specific comments for each reviewer. We refer to line number **xxx** of the main paper as **Lxxx**. Keywords will be highlighted like `this` or **this**. All additional figures are in the `attached pdf`.

We believe that the revisions and additional experiments we have outlined will greatly enhance our paper's quality and impact, addressing the concerns raised. We hope that these responses and our commitment to improving the work will be positively received.

## **A. Responses to Comments for Contents**
**A.1. Moving Algorithm to Main Text (sxrB, tocg)**
>Reviewers sxrB and tocg suggested moving the algorithms to the main text to improve the clarity of our method. We will move Algorithm 1: Sequential Pairwise Comparison RLHF and Algorithm 2: Root Pairwise Comparison RLHF in the supplementary material to Section 4. SeqRank for a clear explanation of the proposed method. To save space, we will illustrate the common framework of both algorithms and specify the trajectory comparison algorithms for sequential and root pairwise comparison. We believe this will enhance the understanding of the proposed algorithms.

**A.2. Overstatement Regarding Cognitive Load (sxrB, tocg)**
>We appreciate the concern that high cognitive load (L030) may seem like an overstatement. For clarity, we will revise the introduction (L029-L033) as follows:\
_“Despite the significant success of RLHF, conventional pairwise comparison requires a human to remember at least two trajectories to determine a single preference. In this regard, it is essential to develop an efficient comparison method that can obtain multiple preference data from a single human feedback, thus reducing a human’s labeling effort.”_

## **B. Additional Experiments and Analyses**
### **B.1. Additional Analysis for Figure 4 in the main paper (sxrB,f3sA)**
>In Figure 4, we illustrate the average reward per number of feedbacks after convergence in DMControl quadruped walk task. We address two major concerns regarding the results in Figure 4.

**(1) When the number of feedback is limited (=100), why does pairwise comparison outperform sequential and root pairwise comparison? (f3sA-Question3)**
>In Figure 4, we fix the frequency of reward learning as learning the reward at each 30,000 policy training steps for fair comparison among different numbers of feedback. The results in Figure 4 imply that for a small number of feedback (=100), the increased data dependency affects the performance more than the increased number of augmented data. However, as the number of feedback gets sufficiently high, the number of augmented preference data affects the performance more than data dependency. Especially for root pairwise comparison, the performance when the number of feedback is 1000 is higher than the performance of pairwise comparison when the number of feedback is 4000. In practical cases, we can learn the reward function more frequently to improve performance if the maximum number of feedback is as low as 100.

**(2) When the number of feedback is high, why does root pairwise comparison outperform the oracle SAC model? (sxrB-Nitpicks and Suggestions 5)**
>A similar phenomenon is shown in previous work [1] that with a sufficient number of feedback, an RLHF agent performs better than an RL agent trained with true rewards. The authors claim that one possible cause of this outperforming behavior is that the reward learning process assigns positive rewards to all behaviors that are typically followed by high rewards. Numerous studies in RL [2, 3] also indicate that learning dense rewards can improve the overall performance of the agent compared to sparse rewards. We believe the strength of reinforcement learning from human feedback also appears in this result, that we can learn a plausible dense reward that potentially leads the agent to perform better.\
\
While RLHF can outperform the oracle model with a sufficient number of feedback, the proposed root pairwise comparison significantly boosts the speed to approach the oracle performance. This is because the high feedback efficiency of root pairwise comparison helps the agent learn the reward model faster than other methods. We address the convergence rate of the reward model in Section 4.2 on page 6 of the main paper.

`Global Responses to All Reviewers are continued (Part II - V) in the following official comments. Dear reviewers, please let the authors know if Part II - V are not visible. We expect the earliest access time would be Aug 9th 11PM PST.`

[1] Christiano et al., "Deep reinforcement learning from human preferences.", Advances in neural information processing systems, 2017.\
[2] Faust et al., "Evolving rewards to automate reinforcement learning.", Workshop on Automated Machine Learning at International Conference on Machine Learning, 2019.\
[3] Zheng et al., "What can learned intrinsic rewards capture?" International Conference on Machine Learning, 2020.

---

### Decision · Program_Chairs · 2023-09-21

**Decision:**

Accept (poster)

**Comment:**

The meta reviewer went through the paper, reviews, responses and agrees with the majority of reviewers that this paper reached the bar of NeurIPS.